# Lab-on-device investigation of phase transition in MoO$_x$ semiconductors

Xiaoci Liang [1], Dongyue Su[1], Younian Tang[2], Bin Xi [2], Chunzhen Yang [3], Huixin Xiu [4], Jialiang Wang [5], Chuan Liu [1] ✉, Mengye Wang [3] ✉ & Yang Chai [5] ✉

Precise tuning of phase transition material properties enables multifunctional devices for information processing and energy conversion, but controlling on-device phase transitions and monitoring microscopic mechanisms remains challenging. Here, we develop a lab-on-device system for molybdenum oxide to probe operando hydrogenation mechanisms through in situ electrical and spectral characterization with density functional theory calculations, revealing threshold-driven proton dynamics that govern the transition between non-volatile memory operation and catalytic hydrogen evolution. Moderate proton intercalation (flux $< 10^{17}$ cm$^{-2}$) achieves a five-order conductance modulation under ambient conditions via polaron formation and stoichiometric optimization (H/Mo up to 22%, Mo/O approaching ideal ratios), outperforming oxygen vacancy engineering. Beyond this threshold (flux ~$10^{17}$ cm$^{-2}$), intensive proton intercalation triggers electric-to-chemical energy conversion, directly linking proton history to catalytic activity. Leveraging these principles, we achieve nonvolatile electrochemical memory with linear synaptic and accumulative neuronal functionalities, and demonstrate an all electrochemical random-access memory neural network hardware that executes memory-efficient rank-order coding for sparse signals even under noisy conditions.

The intricate mechanisms of phase transition, driven by ion migration and redox reactions within semiconductors, can profoundly alter microscopic properties such as atomic bonding, charge distribution, lattice structure, and band structure[1]. These transitions, in turn, significantly affect macroscopic conductance and electrochemical properties, underpinning various electronic and energy conversion applications[2,3]. For example, phase transitions can be utilized for information storage like electrochemical random-access memory (ECRAM)[4]. Additionally, these phase transition can enhance the efficiency of the conversion of electrical to chemical energy by affecting both adsorption and charge transfer processes[5,6].

However, a significant challenge remains in achieving precise control and real-time monitoring of phase transitions, which is crucial for revealing the underlying microscopic mechanisms. This challenge can be addressed by manipulating the dynamics and amount of proton intercalation, a process that can be tuned by the electric field or current applied to phase transition semiconductors[7]. This approach also bridges the isolated studies in electrochemical memory and electric-to-chemical energy conversion processes, allowing to obtain a cohesive understanding of the shared electrochemistry mechanisms. Molybdenum oxide (MoO$_3$) is a typical phase transition material, exhibiting multifold electrochemical properties with multi-redox

[1]State Key Laboratory of Optoelectronic Materials and Technologies, School of Electronics and Information technology, Sun Yat-Sen University, Guangzhou, China. [2]School of Materials Science and Engineering, Sun Yat-sen University, Guangzhou, China. [3]School of Materials, Sun Yat-Sen University, Shenzhen, China. [4]School of Materials Science and Engineering, University of Shanghai for Science and Technology, Shanghai, China. [5]Department of Applied Physics, The Hong Kong Polytechnic University, Hung Hom, Kowloon, Hong Kong, China. ✉e-mail: liuchuan5@mail.sysu.edu.cn; wangmengye@mail.sysu.edu.cn; ychai@polyu.edu.hk

states ($Mo^{6+} \leftrightarrow Mo^{5+} \leftrightarrow Mo^{4+}$) and notable electrical conductivity changes in response to phase transitions induced by proton migration or intercalation[7,8]. Also, it is a widely used material in optoelectronics devices (e.g., OLEDs, QLEDs, and solar cells), energy storage (supercapacitors, Li-ion batteries), and catalysis (HER, $CO_2$ reduction). Controlling these phase transitions could lead to a fundamental reassessment of the operating principles of existing devices and exploring innovation in semiconductor technology.

In this study, we modulate the phase transition of $MoO_x$ through the regulation of proton flux driven by electric field. We investigate the morphological and spectral evolution of $MoO_x$, pre and post proton intercalation, revealing microscopic mechanisms and their impacts on electrochemical memory and electric-to-chemical energy conversion regimes. When a proton flux is $10^{15}$ to $10^{16}$ cm$^{-2}$, proton intercalation enables stable proton adsorption on $MoO_x$'s oxygen, inducing increased carrier density and altered charge transport paths. This enables a nonvolatile increase in conductivity starting within microseconds, reaching a change of five orders of magnitude. When a proton flux exceeds $10^{17}$ cm$^{-2}$, the proton-intercalated $MoO_x$ film undergoes electric-to-chemical energy conversion in electrocatalysis. The enhanced conductivity facilitates faster electron transfer and approximately doubles the reaction rate. These mechanisms reveal the ideal Mo-to-O stoichiometric ratio for broadening the conductance change range and the proper proton flux for stable memory operation. The understanding stimulates the applications on all-ECRAM neuromorphic hardware in processing sparse signals with memory-efficient rank-order-coding.

## Results
### Multidimensional analysis for phase transitions of $MoO_x$
The phase transition of $MoO_3$ is governed by the electric field and proton flux $Q_H$ ($Q_H = \int j_H/q\,dt = \int S_H\,dt$, with $S_H$ being the flux density, $q$ being the elementary charge, and $t$ being the time) within a $MoO_3$-electrolyte-electrode configuration (Fig. 1a). Under the electric field, the proton flux overcomes energy barriers in $MoO_3$-electrolyte interface and the bulk of electrolyte and $MoO_3$. At low $Q_H$, potential barriers at the $MoO_3$-electrolyte interface impede proton injection into $MoO_3$. With a moderate $Q_H$, protons are permitted to penetrate and intercalate into the $MoO_3$ lattice, initiating a phase transition marked by a conductance shift. Upon exposure to high $Q_H$ and $S_H$, the excess protons within the phase-transitioned $MoO_3$ participate in electrochemical reactions, leading to hydrogen gas evolution, a hallmark of the energy conversion process. By gradually increasing the proton flux, we can modulate the on-device phase transition of $MoO_3$, transitioning from memory regime to the energy conversion regime. This approach integrates the two regimes and functionalities within a single semiconductor, with a threshold of proton flux distinguishing between them, thereby enhancing our understanding of the electrochemical reactions and phase transitions.

Regular $MoO_x$ films were fabricated by atomic layer deposition (ALD), forming layered crystalline structures that facilitate proton intercalation. The stoichiometry was controlled by the annealing atmosphere, either in ozone or vacuum. Protons were driven into the $MoO_x$ by an electric field from the protonic electrolyte Nafion, which was deposited on the $MoO_x$. The chemical composition and valence band spectra were investigated by X-ray photoelectron spectroscopy (XPS) analysis for regular $MoO_{2.96}$, and electric-driven proton intercalated $H_{0.22}MoO_{2.96}$ (Fig. 1b). The stoichiometric ratio is calculated according to the proportion of Mo with different valence states[9]. The subpeak positions[10–14] and atomic percentages of Mo3$d$ (Supplementary Table S1) indicate that 92% of Mo exists in the +6 oxidation state in $MoO_{2.96}$, whereas proton intercalation results in a 30% reduction of Mo to the +5 oxidation state in $H_{0.22}MoO_{2.96}$. The O1$s$ peaks (Fig. 1b, right) detect metal-oxygen bonds (M − O), oxygen vacancy (O$_v$) and hydroxyl groups (M-O-H)[15,16]. A pronounced M-O-H peak in $H_{0.22}MoO_{2.96}$

confirms that proton intercalation mainly involves the formation of O-H bonds. A broad peak around 2 eV below the Fermi level in $H_{0.22}MoO_{2.96}$ suggests electron filling in the Mo4$d$ band (Supplementary Fig. S1)[17,18]. XPS analysis reveals that vacuum annealing mainly increases O$_v$ ($MoO_{2.64}$, Supplementary Fig. S2) and induces the formation of M-O-H to a much lesser extent compared to electric-driven proton intercalation. To probe the electronic states, synchrotron radiation X-ray absorption spectra were measured (XAS, Supplementary Fig. S1). The peaks symbolize O2$p$ to Mo4$d$ orbital hybridization (labeled a–c) and O2$p$-Mo5$sp$ hybrid states (labeled d-e) in regular $MoO_{2.96}$. After proton intercalation, peaks a and b are reduced due to the reduction of Mo$^{6+}$ [19], and peak c is diminished, confirming the electron filling of O2$p$ and Mo4$d$ hybrid orbitals and indicating an expansion of the $sp$ state[20,21].

Lattice changes were analyzed using glancing incidence X-ray diffraction (GIXRD) and high-resolution transmission electron microscopy (HRTEM, Fig. 1c,d). The GIXRD patterns confirm the formation of the orthorhombic $MoO_3$[22,23] and the peak shifts induced by electric-driven proton intercalation is aligning with the orthorhombic $H_{0.31}MoO_3$[24–26]. Determined lattice constants indicate an elongated b parameter (interlayer distance) and a marginally modified (110) plane constant (Supplementary Table S2). These results are in good agreement with theoretical values for $HMo_4O_{11}$ in the DFT calculation (Supplementary Table S5). HRTEM images showed a minor change in (100) plane[27,28] after proton intercalation, also approaching theoretical values.

Figure 1e and supplementary Note 2 reveal the energy bands and density of states (DOS) for regular $MoO_3$ and proton-intercalated $HMo_4O_{12}$. In $HMo_4O_{12}$, a transition from low to high carrier concentration, signified with a narrowed band-gap and Fermi-energy within conduction band, and the emergence of a delocalized state at the conduction band minimum are observed, indicating intrinsically tunable conductivity through proton intercalation. According to the d-band center theory[29], the reduced $e_g$ state of Mo4$d$ bonds in proton-intercalated $HMo_4O_{12}$ compared to $MoO_3$ suggests weakened M-O bonds, while the increased DOS of $p_x$ and $p_y$ states suggests strengthened O-H bonds. Bader charge analysis shows that adsorbed hydrogen loses an electron and the oxygen in O-H bonds gains about 0.75 electrons each, enhancing conductance through increased valence charge on oxygen.

We performed theoretical calculations of the partial charge density at the bottom of the conduction band and near the Fermi energy for both $MoO_3$ and $HMo_4O_{12}$, as illustrated in Fig. 1f and Supplementary Fig. S12. After proton intercalation, a notable increase in charge carrier density and a substantial rearrangement in the spatial distribution of charges are observed. In $MoO_3$, conductive pathways emerge through the hybridization of Mo4$d$ and O2$p$ orbitals at symmetric sites (O$_s$), whereas in $HMo_4O_{12}$, they arise from the hybridization at asymmetric sites (O$_a$). These observations suggest a transformation in the charge carrier transport dynamics subsequent to the proton intercalation of $MoO_3$. Also, combining with the lattice analysis results (Fig. 1c, d), these findings suggest, while interlayer VdW forces along the b-axis may be weakened, the change in lattice strain is negligible in the intra-layer direction where charge transport predominantly occurs, implying benefits for device applications due to structural integrity and stability.

### Phase transitions bridging memory and energy conversion
Our investigation into the on-device phase transition started with the fabrication of three-terminal ECRAM devices (Fig. 2a). The phase transition in ECRAM was observed with a moderate proton flux (approximately from $1 \times 10^{15}$ to $5 \times 10^{16}$ cm$^{-2}$), utilizing a solid-state protonic electrolyte Nafion. Protons migrate from the Nafion to the $MoO_x$ interface, undergoing a charge transfer reaction and diffusing within the $MoO_x$, akin to the initial steps of electrocatalysis but without

hydrogen evolution. The electrochemical reaction is represented as:

$$nH^+ + ne^- + MoO_x \rightleftharpoons H_nMoO_x \tag{1}$$

By adjusting the gate current, we could precisely control the proton flux density and, consequently, the degree of proton intercalation in $MoO_x$. Notably, a high gate current of $5\,\mu A$ results in a conductance modulation ratio $G_{max}/G_{min}$ exceeding $10^5$ (Fig. 2b). The retention time for distinct storage states surpasses $1000\,s$, showing good nonvolatility with an ultra-low coefficient of variation $c_v$ ranging from 0.004 to 0.057 (Fig. 2c). The $c_v$ is defined as $c_v = \sqrt{\sum_{i=1}^{l}(G_i - \bar{G})^2/(l-1)}/\bar{G}$, where $G_i$ is the conductance at different times and $\bar{G}$ is the average conductance. Compared with prior data (see Supplementary Table S3 for performance benchmarking), our

devices, fabricated using precise and scalable ALD, achieve a compatibility with existing semiconductor manufacturing processes, and balanced combination of high-performance metrics at ambient conditions, including wide dynamic ranges, long retention times, and good nonvolatility. These advantages, combined with $MoO_x$'s widespread use in optoelectronics (see Supplementary Table S4), underscore the importance of its control on proton content in devices. It is noted that with the same gate current applied for a longer duration or at a higher gate current, bubble formation in ECRAM indicates that the reaction has transitioned into the electric-to-chemical energy conversion regime with the hydrogen evolution reaction (HER), starting at the threshold of the accumulated proton flux of $\sim10^{17}\,cm^{-2}$ at the speed of $3 \times 10^{15}\,cm^{-2}/s$ (Supplementary Fig. S3).

To probe the electric-to-chemical energy conversion mechanism, we developed on-chip electrocatalytic devices that promote reactions such as the HER through applied potentials, which facilitate reactant

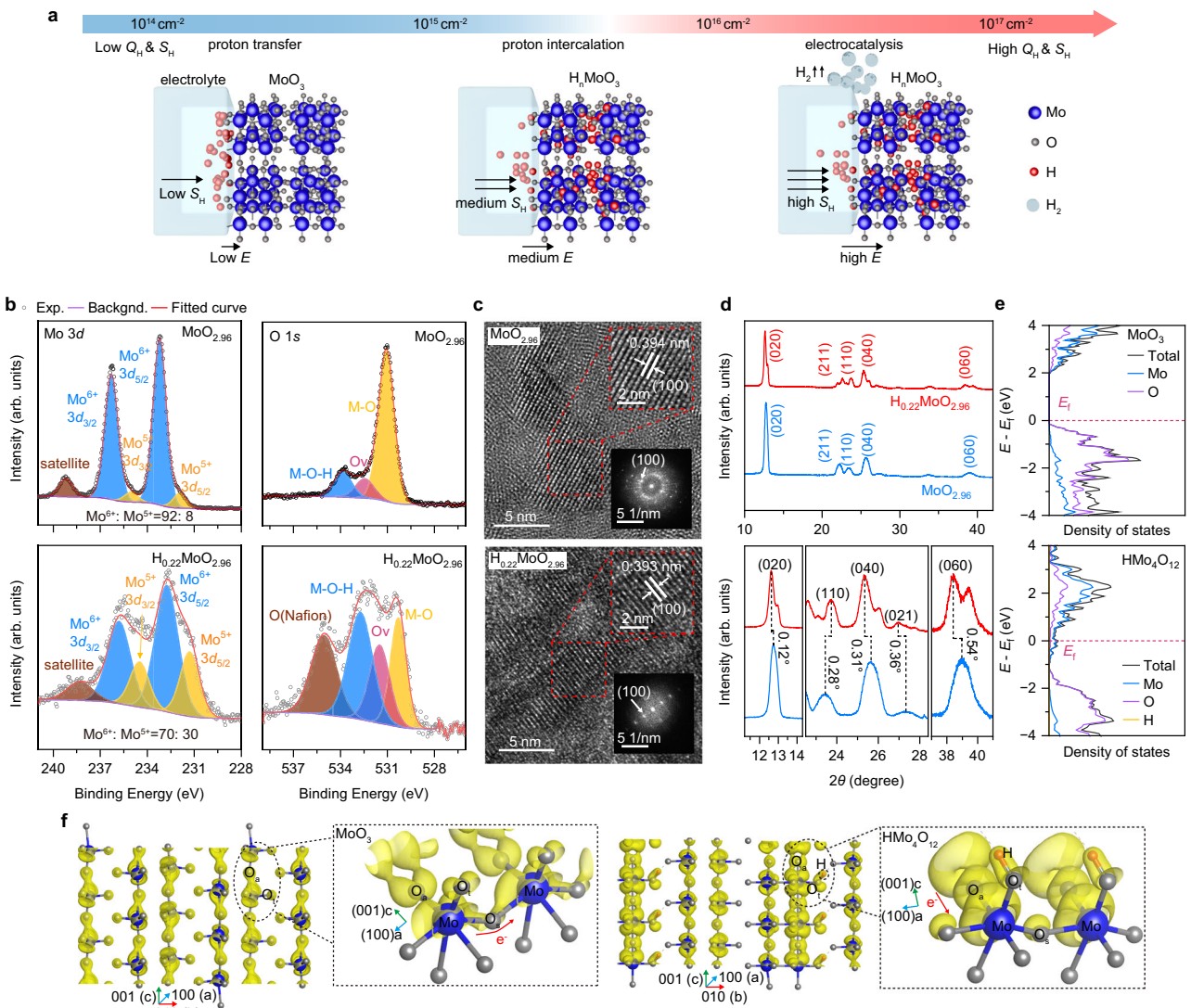

**Fig. 1 | On-device phase transition of $MoO_x$. a** Schematic of the evolution of $MoO_3$ under various total proton flux $Q_H$. Left: at low $Q_H$, protons transfer without reactions; middle: at medium $Q_H$, protons transfer and intercalate into the $MoO_3$; right: at high $Q_H$ and $S_H$, hydrogen gas is generated with the excess protons. **b** XPS corelevel spectra of Mo3$d$ (left) and O1$s$ (right) for $MoO_{2.96}$ (top), and $H_{0.22}MoO_{2.96}$ (bottom). The dot and purple curve represent the experimental (exp.) data and background (backgnd.) data, respectively. **c** High-resolution transmission electron microscopy (HRTEM) images of $MoO_{2.96}$ (top) and $H_{0.22}MoO_{2.96}$ (bottom), with

insets showing zoomed-in views and corresponding fast Fourier transform (FFT) images. **d** X-ray diffraction (XRD) showing crystal structure variation between $MoO_{2.96}$ (blue) and $H_{0.22}MoO_{2.96}$ (red). Enlarged view (bottom) of the (020), (040) and (060) diffraction peaks, highlighting the shift upon electric-driven proton intercalation. **e** Density of states (DOS) for $MoO_3$ (top) and $HMo_4O_{12}$ (bottom) calculated by density functional theory (DFT). **f** The partial charge density isosurface of $MoO_3$ without (left) and with (right) proton intercalation.

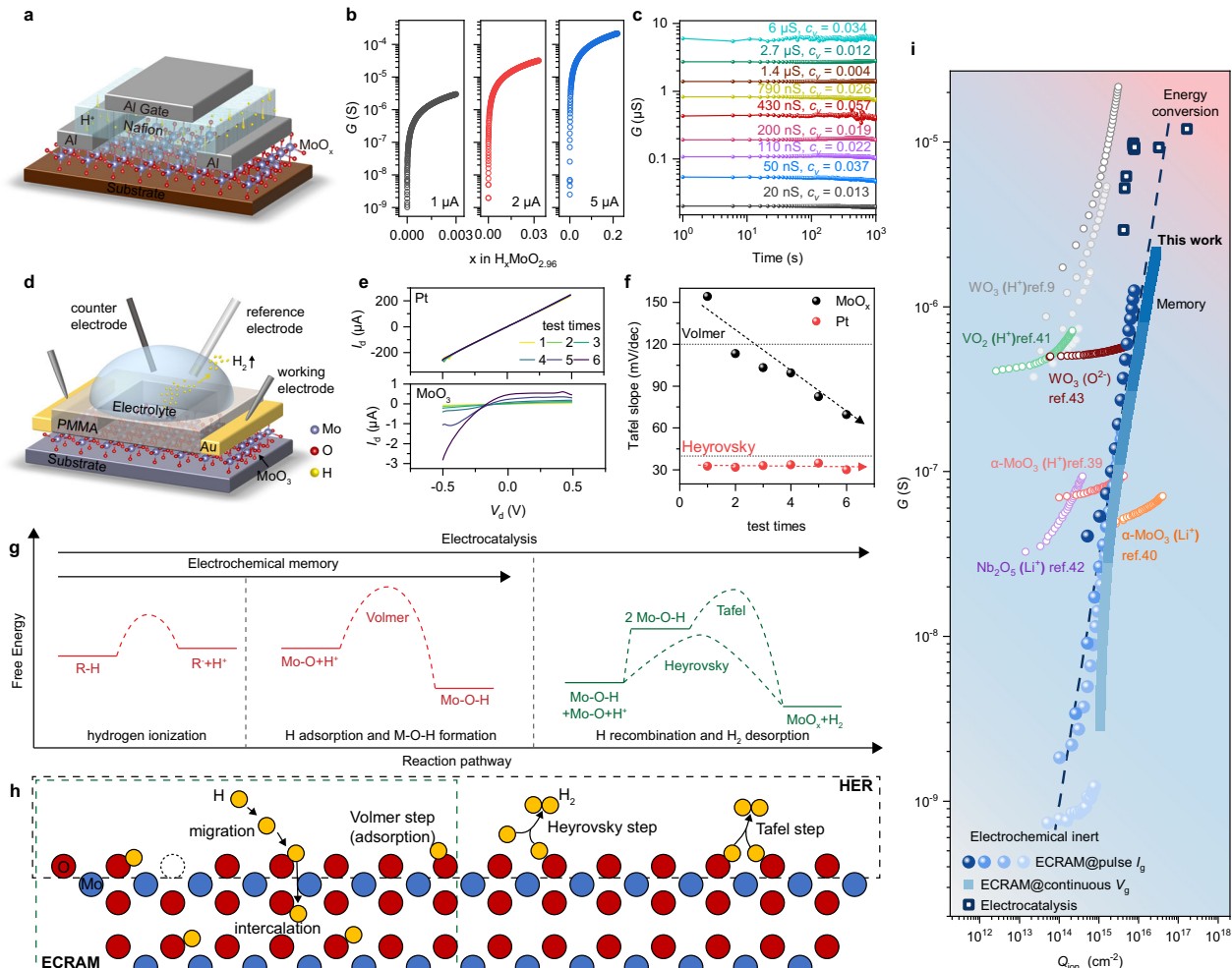

**Fig. 2 | Phase transition correlating electrochemical memory and electric-to-chemical energy conversion. a** Schematic of the ECRAM structure. **b** Conductance modulation under long-term gate current bias. **c**, Retention of 9 programmed analog states ($V_d = 3\,V$) without gate bias, and the corresponding coefficient of variation $c_v$. **d** Schematic of the on-chip electrochemical measurement setup and proposed reactions. **e** The *I-V* curves for Pt (top) and MoO$_x$ (bottom) as a function of the test times. **f** Tafel slopes as a function of the test times. **g** Reaction pathway and free energy diagram for ECRAM and electrocatalysis. **h** The mechanisms of HER and the ECRAM based on proton intercalation, sharing the proton migration, adsorption and intercalation process. **i** Correlation between conductance and proton flux $Q_H$ in ECRAM and electrocatalysis devices, with the dashed line as the guide to the eye. The ECRAM data dots with the same colors were obtained by applying the same amplitude of gate current with various pulse amounts, and the electrocatalysis data dots were obtained from the data in (**e**). Some representative data from previous studies are used for comparison. In the background shadow, light red represents the energy conversion region, light blue represents the memory region, and light gray represents the electrochemically inert region.

adsorption and interfacial charge transfer. Device performance was quantified via standard electrocatalytic metrics including current density and Tafel slope. We used MoO$_x$ serving as the working electrode, H$_2$O as the electrolyte, and a patterned polymethyl methacrylate (PMMA) film acting as a recessed reaction window. An Ag/AgCl electrode functioned as the reference electrode, with a graphite rod serving as the counter electrode (Fig. 2d). An applied bias voltage promotes the adsorption of protons onto the MoO$_x$ surface, initiating the Volmer reaction:

$$H^+ + e^- \rightleftharpoons H_{ads} \tag{2}$$

Such electron transfer from MoO$_6$ octahedra to the H atom can be facilitated by the enhanced conductivity, accelerating the Volmer step. Thereafter, H$_2$ is produced through either the Heyrovsky reaction:

$$H_{ads} + H^+ + e^- \rightleftharpoons H_2 \tag{3}$$

or the Tafel reaction accompanied by electron transfer:

$$H_{ads} + H_{ads} \rightleftharpoons H_2 \tag{4}$$

Interestingly, repeated measurements revealed an increase in MoO$_x$ conductance and a decrease in the Tafel slope to half its original value (Fig. 2e, f), indicating approximately doubling the rate of catalytic kinetics[30–32] and a shift in the rate-determining step from the Volmer to the Heyrovsky mechanism[33–35]. In contrast, a Pt-based electrocatalytic device, which is considered as the HER benchmark[36–38], showed negligible changes in conductance, highlighting the distinctive response of MoO$_x$ due to phase transition. This on-device study reveals that proton intercalation provides a ubiquitous yet unacknowledged continuous assistance in the electric-to-chemical energy conversion process of MoO$_x$. Moreover, we demonstrate reversible modulation between electrocatalytic (HER confirmed by bubble formation) and ECRAM regimes in MoO$_x$ devices, achieving conductance switching (Supplementary Fig. S4), which highlights their operational stability and dual-functional versatility.

The comparative analysis of the reaction pathways in ECRAM and electrocatalytic devices uncovers a shared initial stage, where both processes involve proton immigration and adsorption with charge transfer (Fig. 2g). The processes diverge at subsequent steps—ECRAM relies on maintaining an adsorption state to modulate conductance, whereas electrocatalysis requires a balance between adsorption and desorption to optimize hydrogen production (illustrated in Fig. 2h), adhering to the Sabatier principle. By correlating the total proton flux ($Q_H$) from both experiments, a direct relationship between $Q_H$ and the conductance of $MoO_x$ is established within a wide range of proton flux and conductance modulation, as shown in Fig. 2i, which also displays some representative data from previous studies on other transition metal oxides[9,39–43]. A small $Q_H$ corresponds to the electrochemical inert regime. A moderate $Q_H$ yields an intermediate conductance and promotes stable proton adsorption for electrochemical memory, avoiding hydrogen evolution. In contrast, a high $Q_H$, indicative of a large bias voltage driving a high intercalated proton density over time, correlates with high conductance and accelerates reactions with adsorbed protons and the consequent desorption, thus initiating the energy conversion regime. Note that the conductance increase is primarily driven by proton intercalation-induced phase transitions, with $H_2$ generation being a secondary catalytic effect at high $Q_H$ due to excessive proton accumulation. The above three regimes correspond to the data below, near or above the dashed line in Fig. 2i. This insight confirms the critical role of proton flux in modulating the dual functionality of $MoO_x$, bridging the gap between ECRAM and electrocatalysis. Through delicate control of $Q_H$ by tuning electric field or current, the inherent properties of $MoO_x$ can be harnessed to achieve both high conductance modulation and prolonged retention in ECRAM, while also enhancing catalytic performance.

To further understand the threshold-driven functionality switching, we calculated binding energy of different numbers of protons intercalated in $H_xMo_{16}O_{48}$ (Supplementary Fig. S17). As the number of protons continues to increase, the binding energy tends to stabilize until H/Mo = 0.25, close to the value obtained from the experiment (0.22). Also, its corresponding $Q_H$ ($-3 \times 10^{16}$ cm$^{-2}$) is close to threshold obtained in experiment ($-10^{17}$ cm$^{-2}$). The calculation of unstable adsorption of protons well explains the threshold-driven transition from stable memory to hydrogen evolution reaction. In addition, the approach of precisely controlling proton flux can be applied to other transition metal oxides (e.g., $TiO_2$, $VO_2$, $WO_3$), conductive polymers, and two-dimensional materials, which share the critical traits of proton intercalation capability and electrochemical activity.

## Impacts of M-O-H formation and implications for device performance

The formation of M-O-H bonds is pivotal in the pathway shared by memory and energy conversion regime, presumably including two processes: the mitigation and adsorption of protons, and the formation of polarons (Fig. 3a). The adsorption process, which governs the response speed and the extent of conductivity change, were probed by monitoring the electrical properties. The formation of polarons, resulting from the deformation of the surrounding lattice due to Coulombic forces, were observed through the evolution of optical properties.

The response speed of the phase transition was investigated by measuring the conductance under various electric stimulation durations. Figure 3b displays the enhancement in current following a 10 μs $V_g$-pulse at 10 V for the ECRAM device based on $MoO_{2.96}$. As the pulse width is increased from 10 μs–100 ms, the average conductance change ($\Delta G$) per pulse increases significantly, from 2.1 nS–150 nS (Fig. 3c). Considering the minimum G of $MoO_{2.96}$ at ~1 nS, the device's response threshold is estimated to be around 2 μs for the shortest pulse. The conductance switching observed can be potentially enhanced by reducing the device dimensions, given the dimensions of the device—

featuring a 300 μm $MoO_x$ channel length and a 300 nm thick Nafion film. The μs-scale phase transition initiation not only merits thorough investigation into its impact on $MoO_x$-based devices, including OLEDs, transistors, and solar cells where $MoO_x$ functions as a carrier injection layer[44], and in semiconductor opto-electro catalysis, but also presents opportunities for advancing neuromorphic circuit systems.

The pivotal role of M-O-H bonds in the phase transition implies the significant influence of oxygen sites and vacancies on the reaction rate and conductance dynamic range. In the calculation of $Mo_4O_{11}$ with a single proton in an oxygen vacancy ($O_v$) rich cell ($HMo_4O_{11}$), the $O_{t-s}$ configuration is identified as the most probable steady state with an adsorption energy ($E_{ads}$) of −2.485 eV. For the $HMo_4O_{12}$, the $O_{t-t2}$ configuration is the probable steady state with an $E_{ads}$ of −2.751 eV (Fig. 3d and Supplementary Note 2). The $E_{ads}$ at both $O_a$ and $O_s$ sites show a consistent trend, suggesting a more stable proton adsorption in $HMo_4O_{12}$ than that of $HMo_4O_{11}$. Considering that the electrocatalysis necessitates a balanced adsorption and desorption energy, the high adsorption energy in $HMo_4O_{11}$ may suggest the low energy barrier for desorption with a high reaction rate. The impact of $O_v$ was experimentally investigated through sampled-current-voltammetry of a $MoO_x$-Nafion-metal structure, by sampling the current stimulated by a constant bias at 20 s in the $I$-$t$ curves (Supplementary Fig. S8). Compared with $MoO_{2.96}$, the $O_v$-rich $MoO_{2.64}$ sample exhibits a pronounced saturation behavior at 1.5 to 2 V, reaching the diffusion-limited regime at a lower voltage, evidencing a higher reaction rate (Fig. 3e). We fabricated ECRAM devices using both regular $MoO_{2.96}$ and $O_v$-rich $MoO_{2.64}$. The $MoO_{2.64}$-based device exhibits a higher G due to an enhanced carrier concentration, a larger average $\Delta G$ per pulse, and a lower maximum-to-minimum conductance ratio $G_{max}/G_{min}$ (Fig. 3f, Supplementary Figs. S5-S7). Simulation and experimental results collectively suggest that controlling $O_v$ is key to managing the phase transition and, in particular, the dynamic range of $MoO_x$'s conductance variation broadens as the Mo-to-O ratio approaches the ideal stoichiometric ratio.

The polaron formation, associated with M-O-H bonds, was examined through the induced electrochromic effect, as observed by in-situ real-time optical absorption spectroscopy (Fig. 3g). Under the bias at top electrode, an increase in absorption from 650 nm–720 nm is observed within seconds (Fig. 3h). This observation indicates that, the phase transition, involving the reduction of $Mo^{6+}$ to $Mo^{5+}$, is accompanied by small polarons. The electrons on the $Mo^{5+}$ site can be excited to the $Mo^{6+}$ site through the Franck-Condon transition after photon absorption, resulting in the electrochromic effect. The polaron binding energy $U$, extracted by analyzing the increased adsorption peak[45,46], exhibits a slight decrease over time (Fig. 3i), potentially due to a slight reduction in the interatomic distance between the $Mo^{5+}$-O-$Mo^{6+}$ pair[47–49]. Considering the shorten lattice constant along the a-axis after proton intercalation, these findings suggest that the phase transition induces small polaron within the intra-layer plane (the ac plane) and also indicate that optical properties could be finely tuned alongside conductance changes.

While proton intercalation is known in transition metal oxides like $WO_3$, $VO_x$, and $TiO_2$, the above results advance in several aspects: (1) Achieving broad control over phase transitions in terms of H-to-Mo ratio from nearly 0% to 22%, reduction of $Mo^{6+}$ from 92%–70%, through precise tuning of $Q_H$, resulting in a conductance modulation range from $10^{-9}$ S to $10^{-4}$ S (Fig. 2i). Crucially, DFT calculations and experiments agree well on the threshold-driven memory-to-catalysis transition. (2) Introducing stoichiometric design principles that suppress $O_v$-induced leakage by controlling the Mo-to-O ratio. (3) Uncovering the mechanistic basis of these phenomena through detailed analyses of charge transport, electronic structure changes and the operando hydrogenation dynamics. These contributions providing new insights for developing transitional metal-oxide based electrochemical systems.

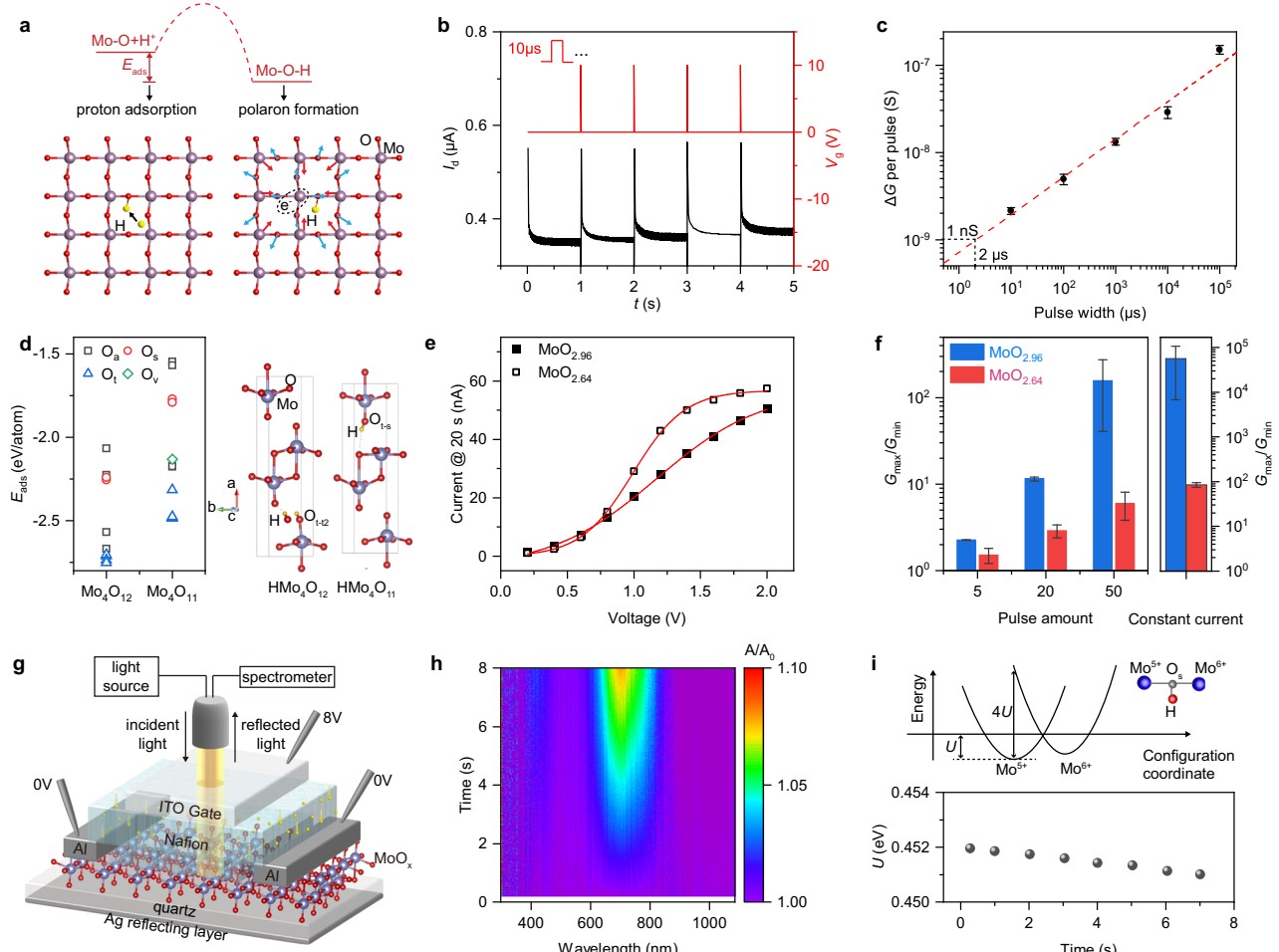

**Fig. 3 | M·O·H formation and implication for device performance. a** The proton adsorption reaction pathway influenced by adsorption energy ($E_{ads}$), and polaron formation. **b** Continuous readout current ($I_d$) update after programming by gate voltage pulses (10 V, 10 μs). **c** Average change in conductance ($\Delta G$) per pulse as a function of pulse width at an amplitude of 10 V. **d** Comparison of proton adsorption energy on $Mo_4O_{12}$ and $Mo_4O_{11}$ (with oxygen vacancies) at various sites. **e** The sampled-current-voltammetry for the $MoO_x$-Nafion-metal structure with or without intentionally induced oxygen vacancies. **f** Comparison of switching ratios for different pulse amounts (left) and the switching ratios for the constant current (right). **g** Schematic of experimental setup used for in-situ real-time optical absorption measurement. **h** The time-resolved UV–vis absorption mapping. **i** The polaron binding energy $U$ evolution. The error bars denote the data in Fig. 3c extracted from 15 measurements and presented as mean values ± standard deviation. The error bars denote the data in Fig. 3f extracted from three measurements and presented as mean values ± standard deviation.

Also, the above studies indicate the significance of integrating memory and catalysis in identifying the threshold and obtaining cross-disciplinary insights. Through the shared mechanism (Fig. 2h), strategies from catalysis, such as defect engineering, surface reconstruction and photo-activation, can likely improve ECRAM performance, while ECRAM principles may also inspire new approaches to enhance catalytic efficiency. This dual functionality extends $MoO_x$'s utility beyond traditional understandings. For example, figures of merit like external quantum efficiency (EQE) in LEDs and Tafel slope in catalysis have been reported without considering how progressive hydrogenation in $MoO_x$ persistently modulate both memory retention and catalytic pathways. According to the above study, device performance is directly tied to proton accumulation history and, therefore, figures of merit in $MoO_x$-based systems needs further investigation under progressive hydrogenation conditions.

## All-ECRAM Synaptic and Neuronal Neuromorphic Network

In the neural network model, synapse and neurons are two basic structures with distinct roles in signal processing. Synapses linearly respond to incoming signals (pre-synaptic impulses), transmitting weighted stimuli (post-synaptic impulses) to the neuron. Neurons integrate these weighted stimuli, which may arrive at different times, and accordingly change the membrane potential which serves as the neuronal output. Conventional hardware-implementation of neural networks have often relied on separately fabricated devices to simulate synapses and neurons[50,51]. These approaches unavoidably involve complex processes to achieve integration of the disparate devices, thereby increasing the technical difficulty of the solutions. The ECRAMs, on one hand, linearly modulate the input signals ($V_{ds}$) into channel currents ($I_{ds}$); on the other hand, it accumulates changes through proton intercalation in response to gate pulses. These device behaviors empower the $MoO_x$ ECRAM to act as a unified device performing both synaptic and neuronal functions in circuits. We demonstrate an all-ECRAM hardware solution designed to simplify the structural complexity of advanced neuromorphic networks.

Leveraging mechanism studies, we employed $MoO_{2.96}$ ECRAM (near ideal in stoichiometry) for its expanded dynamic range and an optimized proton flux $Q_H$ at ~$10^{15}$ cm$^{-2}$ to precisely regulate conductivity, ensuring the discrimination and accuracy of conductivity states with good linearity and symmetry. To mimic intelligent light sensing or memory functions, a commercial photodetector ($\lambda = 700$ nm) was interfaced with the gate terminals of the $MoO_x$

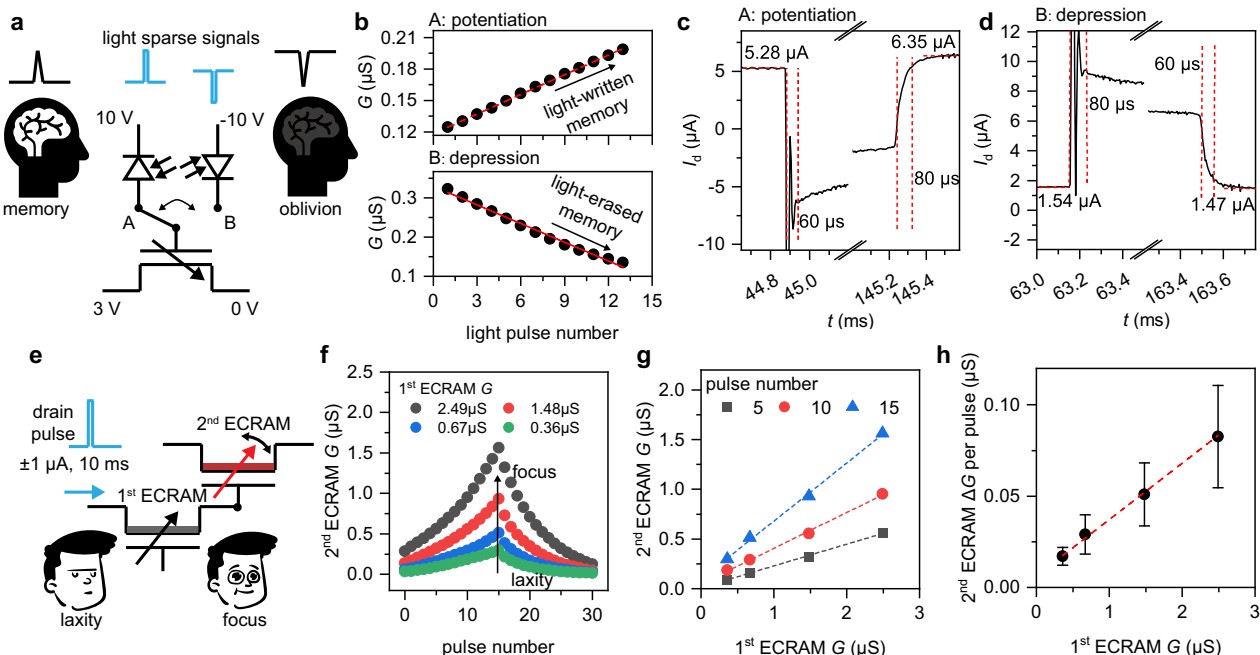

**Fig. 4 | Integration of MoO$_x$ ECRAMs for light and electric response. a** Circuit schematic of ECRAM with photodetector at gate electrode. **b** Programming conductance plotted against number of light pulses. **c**, **d** The rise time and fall time in the potentiation and depression. **e** Circuit schematic of two connected ECRAMs. **f** Programming conductance of 2$^{nd}$ ECRAM across various conductance levels of 1$^{st}$ ECRAM. **g** The conductance of 2$^{nd}$ ECRAM as a function of the conductance of 1$^{st}$ ECRAM. **h** The average change of conductance ΔG of 2$^{nd}$ ECRAM as a function of the conductance of 1$^{st}$ ECRAM. The error bars denote the data in Fig. 4h extracted from 15 measurements and presented as mean values ± standard deviation.

ECRAM (Fig. 4a), which modulated the conductance $G$ in a bidirectional manner. In a forward configuration (Fig. 4b top, with a 10 V bias), $G$ increases from 0.12 μS–0. 20 μS after 12 light pulses, effectively writing light-induced memory[52]. Conversely, when switched to a reverse configuration (Fig. 4b bottom, with a −10 V bias), $G$ drops from 0.32 μS–0.13 μS, erasing the memory through light. The fast phase transition ensures the erasing and writing speed is comparable with the photodetector (Fig. 4c,d, Supplementary Figs. S21–23). This versatile spike-triggered operation of light-written or erased memory has also been accomplished using an organic photodetector in our experiments (Supplementary Fig. S24).

Emulating the human memory process, where memories are reinforced or diminished based on focus or distraction levels, involves integrating multiple MoO$_x$ ECRAMs to modulate memory enhancement or attenuation with spike signals (Fig. 4e, f). When the $G$ of the 1$^{st}$ ECRAM increases from 0.36 μS–2.49 μS, the conductance of the 2$^{nd}$ ECRAM varies linearly at different pulse number (Fig. 4g), and the average change in $G$ (ΔG), stimulated by identical pulses, escalates from 0.017 μS to 0.083 μS (Fig. 4h).

This integrative approach allows for the construction of neural networks capable of processing spatiotemporal signals, such as sequential binary or pulse inputs (Fig. 5a). By expanding into a multi-input matrix, the dual-stage device configuration mimics the biological process where processed action potentials from multiple synapses are integrated by neurons. The 1$^{st}$-stage ECRAMs function as synapses, linearly modulating the incoming sparse pulse signals with the synaptic weights emulated by the channel conductance. The 2$^{nd}$-stage ECRAMs, acting as neurons, integrate weighted signals arriving at different times from the synapses and generate the appropriate enhancement or attenuation of the channel conductance as the neuronal output.

This architecture establishes a foundation for a hardware-implemented spiking neural network (SNN) designed for pattern recognition through temporal rank-order-coding (Fig. 5b, c). Although

conceived as a low-power and memory-efficient solution, the coding algorithm has been limited to software implementations due to the absence of suitable hardware[53]. The developed ECRAMs now bridge this gap by manipulating spatiotemporal pulse signals to adjust the conductance ($G$) of the 1$^{st}$-stage ECRAMs, representing network weights, and programming the conductance change (ΔG) in the 2$^{nd}$-stage ECRAMs after pulse integration (Fig. 5c). During hardware experiments, we performed classification of the characters "o", "J", and "Y" amidst two types of noise: pixel flipping and random grayscale noise. Pixel flipping noise inverts the white and black colors, while grayscale noise results in increased or decreased grayscale (Supplementary Note 4). Unlike conventional ECRAM arrays that perform vector-matrix multiplication for inputs and weights[54], the pulse signals from noisy images are converted into spiking timings through temporal rank-order-coding, where higher grayscale pixels lead to delayed pulse arrivals (Fig. 5c). To synchronize input pulses with variable delays, a custom-designed output pulse unit, governed by a microcontroller unit (MCU), was developed (Supplementary Fig. S27). Employing this coding strategy, nine pulses, acting as the input layer, are fed into the drain electrodes of the 1$^{st}$-stage ECRAMs. The conductance of these ECRAMs serves as the weights. The accumulated pulses $P_j$ are then injected to the gate electrodes of the 2$^{nd}$-stage ECRAMs, inducing ΔG and changes in output current, which function as the output layer for comparative analysis to classify the input images. The mean $G_j/G_0$ ratio, based on the summed pulse amplitude, linearly increases with amplitude (Fig. 5d), ensuring the accuracy needed to classify noisy images.

As illustrated in Fig. 5e, an example classification of a noisy "o" image is demonstrated. The 'o' channel exhibits the highest pulse signal integration, resulting in a 16% increase in conductance, outperforming the 'J' and 'Y' channels with 12% and 7% increases, respectively. The weights of the 1$^{st}$-stage ECRAM were trained in software (see the flowchart in the Supplementary Fig. S26), with learning rules adapted from error backpropagation to account for spike latencies[53].

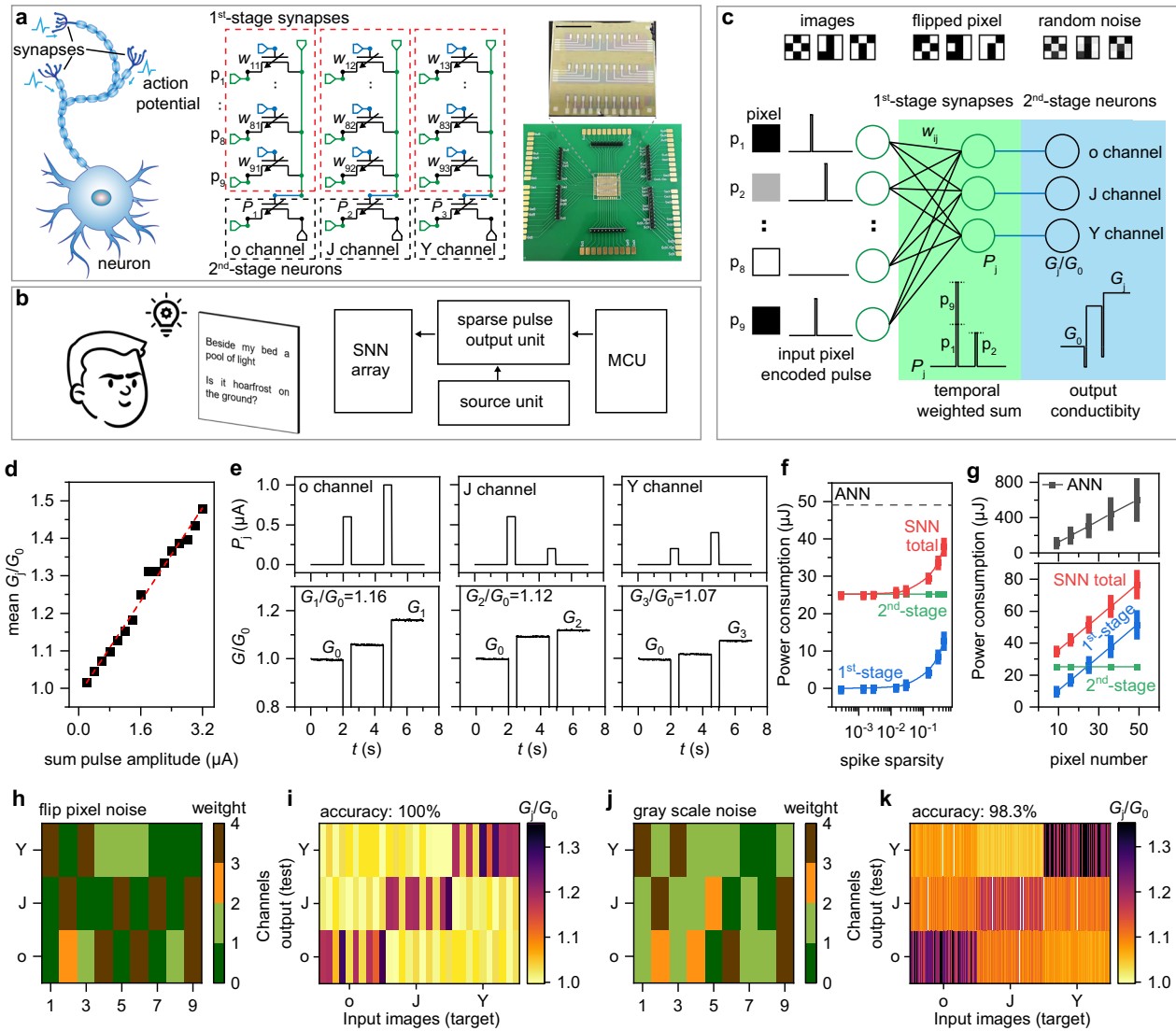

**Fig. 5 | Implementation of the spiking neural network. a** Left: signals from the synapses are integrated and processed by the neurons; Middle: circuit diagram of a 2-stage SNN unit; Right: the optical image of an SNN hardware by integrating 30 $MoO_x$ ECRAM devices. **b** Implementation of letter recognition through human or hardware with SNN algorithm. In hardware, input pulses with variable delays are governed by the microcontroller unit (MCU). **c** Noisy image examples and schematic of the SNN architecture. **d** Average $G_j/G_0$ plotted against total pulse amplitude. **e** $P_j$-$t$ and $G/G_0$-$t$ curves for three channels, demonstrating recognition of a

noisy "o" image. **f, g** Estimated power consumption for the spiking neural network (SNN) and artificial neural network (ANN) with various spike sparsity (**f**) and pixel numbers (**g**). **h** Normalized weight distribution for flipped pixel noise conditions. **i** $G_j/G_0$ as the output for the input various images with flipped pixel noise. **j, k** Corresponding weight distribution and output $G_j/G_0$ for gray scale scenarios. The error stick in (**j**), (**k**) was evaluated according to the variation of device conductance after programming.

The weights are normalized, evenly divided into four values, and then scaled to be compatible with hardware execution, as shown in Fig. 5h and j. The hardware implementation results are shown as a heatmap in Fig. 4i and k, which displays the $G_j/G_0$ values from the three channels as the output for the input various input "o", "J", "Y" images. The system achieves a 100% accuracy rate for flipped pixel images and a 98.3% rate for noisy grayscale images, showing robustness in handling complex pattern recognition tasks. Our SNN, utilizing rank-order-coding, efficiently processes inputs with a single spike and stores inference results, reducing power consumption compared to spike-count-based-coding SNNs, which require multiple spikes. Also, compared to an artificial neural network (ANN)[55,56] operating under the similar structural framework but with continuous current signals, the SNN's use of sparse signal pulses results in lower power consumption per inference, as estimated in Fig. 5f and supplementary Note 4. This efficiency is sustained with increased spike sparsity for enhanced classification

accuracy or with higher pixel counts (Fig. 5g). Overall, the experimental execution confirms the system's robustness and efficiency in pattern recognition under various conditions.

## Discussion

Our study fundamentally advances the understanding of proton-driven phase transitions in $MoO_x$, establishing design principles that bridge electrochemical memory and electrocatalysis. At the core of this advance is the identification of a proton flux threshold ($Q_H \sim 10^{17} cm^{-2}$, H/Mo ~ 22%), which acts as a switch between memory and catalytic functionalities within a single material platform. Below this threshold, proton intercalation induces nonvolatile conductance modulation through polaron formation, enabling conductance changes within microseconds. Above it, excessive proton accumulation triggers hydrogen evolution reactions, remarkably enhancing catalytic activity. Such threshold-driven proton dynamics are resolved by in situ

electrical and spectral observations and DFT calculations, revealing the operando hydrogenation mechanisms of how proton accumulation history changes both memory retention and catalytic activity. Crucially, stoichiometric control demonstrates that precise Mo/O ratios outperform traditional oxygen vacancy engineering in balancing dynamic range and stability. This paradigm shift enables five-order conductance modulation while maintaining ambient operation. Finally, the interplay between these mechanisms is harnessed in an all-$MoO_x$ ECRAM neuronal neuromorphic network, where proton history-dependent conductivity naturally implements rank-order coding. This architecture mimics brain-like responses to signals and achieves high-precision image classification, even under noisy conditions. By unifying electronic and electrochemical functionality through proton dynamics, this study reveals that the widely used $MoO_x$-based system is dynamically reconfigured by transient proton fluxes. These findings may inspire further exploration of $MoO_x$'s role in opto-electronics, electrochemical memory, smart catalysis, and beyond.

## Methods

### Film deposition

First, $MoO_3$ was deposited by atomic layer deposition (ALD) on the 100 nm $SiO_2$/Si substrate and in situ annealed to obtain crystalized orthorhombic $MoO_3$ phase, which was marked as $MoO_{2.96}$ according to the XPS result. All ALD processes were performed in a commercial PICOSUN R-200 Advanced type reactor. The ALD precursor source is $Mo(=N^tBu)_2(CH_2SiMe_3)_2$ and the deposition temperature was set at 300 °C. The growth per cycle was determined to be 0.60 Å/cycle, and a total of 1000 cycles yielded a film thickness of ~58 nm, as measured by using an Ellipsometry (J. A. Woollam Co. alpha-SE). After deposition, the films were annealed at 400 °C in ozone or in vacuum for 15 min to form $MoO_{2.96}$ or $MoO_{2.64}$, respectively.

### Film characterization

The X-ray photoelectron spectroscopy (XPS, Thermo Fisher ESCALAB 250Xi) measurement was carried out to analyze the chemical constituents and the valence band spectrum using a monochromatized Al Kα source. The stoichiometric ratio of molybdenum oxide is calculated according to the proportion of molybdenum with different valence states. In stoichiometric $MoO_3$, the Mo in the +6 oxidation state is associated with three oxygen atoms. In $O_v$-rich $MoO_x$, the reduced valence state of Mo corresponds to the decrease in the number of oxygen atoms. In proton intercalated $MoO_x$, the reduced valence state of Mo corresponds to the increase in the number of hydrogen atoms.

Glancing incidence X-Ray Diffraction (GIXRD) pattern was collected on D8 ADVANCE with DAVINCI DESIGN (Cu Kα λ = 1.5406 Å) to determine the crystal structure with an angle of 0.5°. Powder diffraction files for $MoO_3$ (ICDD No. 00-005-0508), $Mo_4O_{11}$ (ICDD No. 04-005-4333) and $H_{0.31}MoO_3$ (ICDD No. 01-070-0615) are used as comparison. GIXRD patterns display peaks at 2θ = 12.7°, 23.5°, 25.7°, and 39.0°, corresponding to the (020), (110), (040), and (060) planes of orthorhombic $MoO_3$, and a peak at 2θ = 22.2° attributed to the (211) plane of $Mo_4O_{11}$[22,23], confirming the presence of both phases based on XPS analysis. Electric-driven proton intercalation induces shifts in the (020), (040), and (060) peaks to 2θ = 12.6°, 25.3°, and 38.4°, aligning with orthorhombic $H_{0.31}MoO_3$[24–26].

O K-edge X-ray absorption fine structure (XAFS) was performed at BL11U at Hefei Light Source (Hefei, China). The photon flux and energy resolving power were ~$5 \times 10^{10}$ phs/s and ~15,000, respectively, and the beam size at the sample was set to 200 × 100 μm. The O K-edge was collected using the total electron yield (TEY) mode at room temperature under ultrahigh vacuum chamber ($10^{-9}$ Torr). In the O K-edge XAS of $MoO_{2.96}$ (Supplementary Fig. S1), five key features emerge, with the 530–537 eV range (labeled a–c) mapping to O2$p$ to Mo4$d$ orbital hybridization, while features d and e correspond to O2$p$-Mo5$sp$ hybrid states[57–59]. The $t_{2g}$ and $e_g$ orbitals is originated from the Mo4$d$ and O2$p$

hybrid orbitals split by the octahedral crystal field of oxygen (Supplementary Fig. S13). The electron energy loss spectroscopy data (Supplementary Fig. S1) further corroborates these observations, particularly the subtle alterations in the $e_g$ (reflected in peak c of the XAS) and $sp$ states. The crystal structure was examined by aberration-corrected high-resolution transmission electron microscopy (HRTEM) with a ThermoFisher Spectra 300 microscope operated at 300 kV.

### In situ real-time optical absorption measurement

The molybdenum oxide is deposited on the quartz and Indium-Tin-Oxide (ITO) is deposited as top gate electrode on Nafion. A tungsten halogen was used as the white light source in this measurement. After incident light ($I_0$) went through the sample and reflected by the Ag film, the reflected light ($I_R$) was collected by the spectrometer (Ocean Optics QE pro) with wavelength range of 300–1100 nm and time interval of 0.01 s. Then, absorption spectra were obtained using the following equation: $A(\lambda) = \log(I_0/I_R)$. The corresponding polaron binding energy $U$ is extracted by analyzing the increased adsorption peak at 680 nm by: $A = D(8\pi U h\nu_0)^{-1/2} h\nu \exp[-(h\nu - 4U)^2/(8Uh\nu_0)]$, where $h\nu$ is the photon energy, and $h\nu_0$ is the vibrational phonon energy[45,46].

### Density functional theory calculations

The first-principles calculations were performed in the framework of the density functional theory with the projector augmented plane-wave method, as implemented in the Vienna Ab initio Simulation Package (VASP)[60]. The exchange-correlations of electrons are described by the generalized gradient approximations (GGA) with the form proposed by Perdew, Burke, and Ernzerhof[61]. The strongly constrained and appropriately normed (SCAN) meta-GGA functional was employed to accurately describe the geometries and the energies[62,63]. The long-range Van der Waals interaction is described by the DFT-D3 approach[64]. The cut-off energy for plane wave is set to 500 eV. The Brillouin zone integration was performed using a 2 × 7 × 6 k-mesh. The converged conditions for electronic and ionic optimizations were respectively chosen as $10^{-5}$ eV and 0.02 eV/Å. Bader charge analysis is utilized to approach the atomic charges and charge transferring in heterostructures[65]. The adsorption energy $E_{ads}$ is computed by: $E_{ads} = (E_{ad/sub} - E_{ad} - E_{sub})/n_H$, where $E_{ad/sub}$, $E_{ad}$, $E_{sub}$, and $n_H$ are the total energies of the optimized adsorbate/substrate system, the adsorbate in the structure, the clean substrate, and the number of hydrogen atoms, respectively. In this work, the clean substrate represents the $MoO_3$ or $Mo_4O_{11}$, whereas the adsorbate indicates the hydrogen. To simulate the systems with various hydrogen concentrations, we selected a $Mo_{16}O_{48}$ supercell with $n_H$ ranging from 1 to 8 (maximum H/Mo ratio: 50%), containing 40 possible positions for each H atom. Using the Structures of Alloy Generation and Recognition (SAGAR)[66] method, we generated candidate structures and employed two biased screening schemes: (1) selection of structures with fewer Wyckoff positions, which will become stable structures with greater probability[67]; (2) gradual generation of higher concentration structures based on stable low-H-concentration structures[68]. As a result, we evaluated the total energies of over 1000 candidates using the first-principles method. The optimized structures and charge density were visualized by VESTA[69].

### Device fabrication and characterizations

For the three-terminals ECRAM device, a transistor-like configuration was adopted. First, $MoO_3$ was deposited by ALD on the 100 nm $SiO_2$/Si substrate at 300 °C and was further in situ annealed. The source and drain electrodes consist of 60 nm thick Al was deposited by vacuum thermal evaporation. A channel with a width of 1000 μm and length of 300 μm between the source and drain electrodes was patterned by shadow mask. The Nafion precursor was repeatedly spin-coated 4 times at a speed of 2000 rpm for 30 s to constitute the solid-state electrolyte layer followed by thermal drying. 100 nm Al top gate

electrode was deposited by vacuum thermal evaporation through a shadow mask. The conductance characteristics were measured using an Agilent B1500A semiconductor parameter analyzer, PDA FS380 and RIGOL DG1032 pulse generator. The gate current pulses with various amplitudes, widths and amounts were used to stimulate the devices. The ECRAM results used to plot the relationship between $Q_H$ and $G$ were obtained by applying the gate current (at 0.5, 1, 2.5, or 5 μA) with various pulse numbers. The error bars denote the data in Fig. 3f and Supplementary Fig. S7c, f extracted from three measurements and presented as mean values ± standard deviation. The error bars denote the data in Figs. 3c, 4h and Supplementary Fig. S6 extracted from multiple pulse measurements (5 or more times) and presented as mean values ± standard deviation.

For the on-chip electrochemical measurement, a patterned electrode with 5 nm Cr and 50 nm Au was deposited on the $MoO_x$ by vacuum thermal evaporation as the working electrode. The poly-methyl methacrylate (PMMA) precursor was spin-coated on the $MoO_x$ film as passivation layer at a speed of 1500 rpm for 40 s. Subsequently PMMA was annealed at 90 °C for 20 min and treated by plasma to expose the reaction window. All on-chip electrochemical measurements were performed using a semiconductor parameter analyzer PDA FS380 in a four-electrode setup with a Ag/AgCl electrode as a reference and a graphite rod as a counter electrode as ref. [5] A droplet of deionized water was placed on the sample. The reference and counter electrodes were immersed in the droplet. The electrocatalytic activity was examined by polarization curves using linear sweep voltammetry at room temperature. The conductance of the film was measured by two Cr/Au electrodes. The proton flux $Q_H$ is calculated using $Q_H = \int j_H/q \mathrm{d}t = \int S_H \mathrm{d}t$, where $j_H$ is proton current density, $S_H$ is proton flux density, $q$ is elementary charge, and $t$ is time. For ECRAM devices, $S_H = I_G/(Aq)$ ($I_G$ is gate pulse amplitude, $A$ is channel area), and $t = t_p \times n$ ($t_p$ is pulse width, $n$ is number of pulses). Thus, $Q_H = S_H \times t = I_G/(Aq) \times t_p \times n$. For example, with $I_G = 1$ μA, $A = 0.003$ cm², $t_p = 50$ ms, and $n = 15$, $Q_H = 1.56 \times 10^{15}$ cm⁻². For on-chip electrocatalytic devices, $Q_H = \int j_H/q \mathrm{d}t = \int I_C/(Aq)\mathrm{d}t$ ($I_C$ is counter electrode current, $A$ is electrochemical reaction area defined by PMMA window).

## SNN hardware

The 30 ECRAM devices for SNN are fabricated on the same substrate. For the connection of the source electrode and gate electrode, the Nafion film was patterned by carefully erased to exposed source electrode before the deposition of gate electrode. For the image classification task, we first train the network on software based on a single-spike supervised spiking neural network. The network contains 9 inputs and 3 outputs. In this network, the image pixels were converted to spike by time-to-first-spike coding. Through delivering and accumulating the spike signal, the spike arrive times of output neurons were used to classified the images. The connection weights between input and output neurons were updated to minimize the loss function by using the stochastic gradient descent and backpropagation algorithms. Two image sets based on 3 × 3 pixels "o", "J", "Y" were used. One set was introduced flip pixel and containing 30 images, the other was introduced 20% random noise in gray scale and containing 999 images. After training, the weights were normalized and divided evenly into four values. Then, the weights were programmed into the ECRAMs. The same image sets were converted into a series of pulses. For the flip pixel noise, all pulse arrive time is the same. For the random noise, the pulse arrives at the first period when pixels with 0%–10% noise and arrives at the second period when pixels with 10%–20% noise. With the pulse input, the conductance of the ECRAMs were measured simultaneously.

## Data availability

The authors declare that the experimental and simulation data supporting the results of this study can be found in the paper and its Supplementary Information file. The detailed data for the study is available from the corresponding author upon request. Source data are provided with this paper.

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

## Acknowledgements

This project was supported by the National Key Research and Development Program of China (2021YFB3600701, C.L.), the National Natural Science Foundation of China (62425405, Y.C., 62404255, X.L.), the National Natural Science Foundation of China (U23B20166, M.W.), Guangdong Basic and Applied Basic Research Foundation (2024A1515030039, M.W.).

## Author contributions

C.L., M.W. and Y.C. designed the project. X.L. and D.S. designed and conducted the experiments, and analyzed the data; Y.T. and B.X. contributed to the $MoO_x$ film deposition; C.Y. contributed to the XAFS characterizations; H.X. contributed to the TEM characterizations; M.W. contributed to theoretical calculations. C.L., M.W., J.W. and and Y.C contributed to data analysis. X.L., C.L., M.W. and Y.C. wrote the manuscript.

## Competing interests

The authors declare no competing interests.
