## [Transparent Peer Review file · Nature Communications]

Lab-on-device investigation of phase transition in MoOx Semiconductors

Corresponding Author: Professor Chuan Liu

Version 0:

Reviewer comments:

Reviewer #1

(Remarks to the Author)

This paper investigated electric-driven proton intercalation in the MoOx devices, to modulate phase transitions for multifunctional device applications in memory storage and energy conversion. The authors constructed a Lab-on-device system, achieving the development of an integrated system to study phase transitions in MoOx, and allowing precise control and monitoring of electric-driven proton intercalation. The major application of such MoOx devices seem to be in electrochemical memristor and neuromorphic devices, yet the title of this paper is constructed in a way as if the mechanistic investigation of the phase transition is the major focus. Overall, this work fails to convince the true novelty and practical significance. First, the proton intercalation is a well-documented phenomenon across a wide range of transition metal oxides. This includes materials such as WO₃, VO_x, and TiO₂, etc., which have been extensively studied for electrochemical-based applications. Given this broader context, the fundamental concept of proton intercalation itself is not inherently novel. Second, for the practical application of memristor and neuromorphic devices, similar applications have been achieved previously with much better performance (MoS₂_xO_x, Nat. Electron. 2018, 1, 130), and the overall performance reported in this work, including ON/OFF ratio of ~10⁵⁻⁶ and retention time of 10³ seconds, are pretty average in the corresponding field. Moreover, what true advantages this neuromorphic device structure can provide, to achieve any specific goals that are difficult to other neuromorphic devices already developed in the field? Thirdly, the authors wrote the paper in a way that seem to claim the importance of integrating both memory and catalysis for dual-functional devices, which I do not agree as these two functions are so different that they do not need to be achieved within a single device setup. Overall, unless the authors can provide detailed performance data and comparison to the state-of-the-art devices, either in terms of catalysis or in neuromorphic devices, to indicate the reported device structure can overperform the corresponding field, I do not recommend the publication of this paper in this prestigious journal.

Reviewer #2

(Remarks to the Author)

The work titled "Lab-on-device investigation of phase transition in MoOx semiconductors" by Xiaoci Liang et al. presented an electrically driven proton intercalation method based on MoOx semiconductors to regulate the electrochemical properties of the material, which holds significant implications for fields such as information processing and energy conversion. The work is interesting, but there are still some issues that need to be addressed.

1. Please reclaim the novelty of this work for publishing in Nature Communications.
2. Why is it necessary to investigate the mechanisms of phase-transition devices based on MoOx? What are the advantages of MoOx as a material? Do phase-transition devices based on MoOx possess specific advantages compared to other phase-transition devices? The article seems to lack comparisons with other works.
3. It would be helpful to provide a brief introduction to electrocatalytic devices within the text to avoid confusion for readers unfamiliar with this field. Additionally, the manuscript compares the MoOx-based electrocatalytic device with a Pt-based electrocatalytic device. Could you clarify why this comparison was chosen? Specifically, what category of devices does the Pt-based electrocatalytic device represent, and compared to neuromorphic devices, what parameters or performance metrics are more emphasized in Pt-based electrocatalytic devices? Adding these clarifications would make the manuscript more accessible to a broader audience and provide clearer context for the comparison.
4. The blue arrows in Figure 3a are too small and could be enlarged to improve visibility and clarity of the illustration.
5. It is suggested to include additional discussions, such as how this technology could be extended to other materials or devices, and how it specifically advances the development of next-generation neuromorphic systems.

Reviewer #3

(Remarks to the Author)

I have read through the manuscript by Liang et al. with great interest. The authors investigate the on-device phase transition of MoO_x semiconductors under electric-driven proton intercalation, demonstrating a unique interplay between electrochemical memory (ECRAM) and electric-to-chemical energy conversion. By precisely controlling proton flux, the authors identify two regimes: moderate flux ($\sim 10^{15}$ - 10^{16} cm⁻²), where stable proton adsorption enables a conductance modulation ratio of 10^5 and excellent memory retention, and higher flux ($>10^{17}$ cm⁻²), where hydrogen evolution reactions occur, transitioning the system to electrocatalytic behavior. The authors elucidate the roles of M-O-H bond formation, oxygen vacancies, and polaron dynamics in these processes through detailed characterizations, including XPS, in-situ absorption spectroscopy, and DFT calculations. Integrating MoO_x ECRAMs into a neuromorphic hardware platform for rank-order-coded spiking neural networks further highlights the practical significance of this work. This work is solid. Therefore, I can recommend this manuscript for publication in Nature Communications after the following changes are implemented.

1. The most critical factor in this study is the proton flux, which determines the operational regime of the material: the ECRAM region at moderate proton flux and the electrocatalyst region at higher flux. A key question that arises is whether the transition between these two regimes is reversible. Specifically, after operating in the electrocatalytic regime under high proton flux, can the system return to the ECRAM regime if the proton flux is subsequently reduced? Clarifying the reversibility of these regimes would provide deeper insights into the material's stability and functional versatility.
2. There seems to be insufficient explanation regarding how the values of proton flux can be calculated during the experiments. A more detailed description of the methodology used to determine proton flux would significantly enhance the clarity and reproducibility of the study.
3. In the description of Fig. 1a, the authors state that for low proton flux (QH), proton injection into MoO₃ is impeded by the potential barrier. Also, proton flux density can be precisely controlled by the gate current. It seems that low QH corresponds to low gate voltage or gate current, which prevents the potential barrier from being crossed. Then, what is the potential barrier? And could the authors clarify why and how proton flux can be directly related to the potential barrier?
4. In Fig. 2h, the authors explain that the conductivity of MoO₃ changes due to a phase transition at moderate QH and increases further due to the generation of H₂ via electrocatalytic reactions at high QH. Could the authors provide more detail about how H₂ generation contributes to the increase in conductance in the high QH regime?
5. On page 9, lines 10–12, the text seems to describe Supplementary Fig. S4 rather than Supplementary Fig. S3. Please revise this section to reference the appropriate figure correctly.
6. In Figure 4, the authors emulate the human memory process by using the output of the 1st ECRAM as the gate input for the 2nd ECRAM, effectively controlling the proton flux. The implementation of rank-order-coded spiking neural networks using this approach is highly impressive. However, the demonstration of the memory function using light in Figure 4a and 4b, described as being utilized for an organic photodetector, appears to be presented alongside the rank-order-coded spiking neural network results. Fig. 4a, b placement may give the impression that the light-induced memory function is directly related to the rank-order-coded network, potentially confusing readers.

Version 1:

Reviewer comments:

Reviewer #1

(Remarks to the Author)

The authors have addressed my concerns, and now I can recommend the publication of this manuscript in this journal.

Reviewer #2

(Remarks to the Author)

The authors have addressed my previous concerns, I have no more questions now.

Reviewer #3

(Remarks to the Author)

The authors have provided comprehensive and well-reasoned responses to all previous comments. The detailed tests conducted on Nafion/MoO_x ECRAM devices clearly demonstrate the reversibility between the electrocatalytic and ECRAM regimes, directly addressing the concern of whether the transition between these two operational states is reversible. The authors presented a detailed explanation of the proton flux calculation method. The authors' discussion on the relationship between proton flux, gate voltage, and the potential barrier is clearly articulated. In addition, they clearly distinguished H₂ generation from the phase transition and conductance change in MoO₃, elucidating the specific mechanism by which proton insertion alters its conductance. Finally, the separation of the organic photodetector and rank-order-coded spiking neural network led to a clearer and more precise presentation of the content. These revisions not only address the earlier concerns

but also enhance the overall quality and impact of the work. I support publication of the manuscript in its current form in Nature Communications.

Point-by-point Response

We would like to thank all reviewers for carefully evaluating our manuscript. We also thank VERY MUCH for the reviewers' time and efforts to provide us detailed and valuable comments. The constructive and professional suggestions are greatly helpful to us for improving the quality of this work. We are delighted that the 2nd and 3rd reviewers speak highly of our work and recommend the publication of this paper after proper revisions. The 1st reviewer concerns the novelty and practical significance of this work. We have addressed all the concerns raised by the reviewers point-by-point as follows.

Reviewer #1 (Remarks to the Author):

This paper investigated electric-driven proton intercalation in the MoO_x devices, to modulate phase transitions for multifunctional device applications in memory storage and energy conversion. The authors constructed a Lab-on-device system, achieving the development of an integrated system to study phase transitions in MoO_x, and allowing precise control and monitoring of electric-driven proton intercalation. The major application of such MoO_x devices seem to be in electrochemical memristor and neuromorphic devices, yet the title of this paper is constructed in a way as if the mechanistic investigation of the phase transition is the major focus. Overall, this work fails to convince the true novelty and practical significance.

First, the proton intercalation is a well-documented phenomenon across a wide range of transition metal oxides. This includes materials such as WO₃, VO_x, and TiO₂, etc., which have been extensively studied for electrochemical-based applications. Given this broader context, the fundamental concept of proton intercalation itself is not inherently novel.

Reply: We appreciate the reviewer's comment regarding the well-documented nature of proton intercalation in transition metal oxides. We agree with the reviewer that proton intercalation has been studied in materials such as WO₃, VO_x, and TiO₂. The contribution of this work mainly lies in the quantitative study on the on-device phase transition for memory and energy conversion functions, which share the phase transition mechanism but requires different proton flux. This quantitative mechanism study identifies the threshold between the two processes, which advances the understanding and application of this phenomenon in MoO_x-based devices.

(1) This study demonstrates **wide range of control over on-device phase transitions**. We demonstrate wide range of control over phase transitions in MoO_x by tuning the proton flux (Q_H) **from 10^{14} to 10^{17} cm⁻², H-to-Mo ratio from nearly 0% to 22%, reduction of Mo⁶⁺ from 92% to 70%.**

Consequently, we achieve a conductance (G) modulation range from 10^{-9} S to 10^{-4} S. As shown in **Figure R1**, we identify a threshold of approximately 10^{17} cm^{-2} that bridges the memory and energy conversion functionalities. Below this threshold, the device exhibits rapid conductance changes within microseconds, which is competitive for electrochemical memory applications. Above the threshold, the device nearly doubles the electric-to-chemical energy conversion rate, highlighting its potential in catalytic applications.

In contrast, previous studies on materials such as $\alpha\text{-MoO}_3$, VO_2 , and Nb_2O_5 exhibit lower conductance modulation ranges under similar ion flux¹⁻⁴, while WO_3 -based devices^{5,6} show a larger conductance ($>10^{-6}$ S) but with a narrower range of ion fluxes (below 10^{16} cm^{-2}), which do not enter the catalytic regime (**Figure R1**). Our work reveals that proton intercalation in MoO_x can induce a broad modulation range of phase transitions and conductance. These results demonstrate the unique value of MoO_x in phase transition control.

Figure R1. The $Q_{\text{ion}}\text{-}G$ relation of the electrochemical device of various materials.

(2) Such performance is enabled by our unique approach regarding **the critical role of stoichiometric control on phase transitions**. We demonstrate that the Mo-to-O stoichiometric ratio in MoO_x significantly affects proton adsorption energy and charge transfer dynamics, especially in delocalizing the intralayer charge transport. By controlling the Mo-to-O ratio (e.g., a high Mo:O ratio of 1:2.96), we not only broaden the conductance on/off ratio but also suppress leakage currents caused by oxygen vacancies. See **Fig. 3f** (the effect of oxygen vacancies on on/off ratio) and **Extended Data Table 1** (the Mo/O stoichiometric ratio data) for the critical role of stoichiometric ratio control.

This *contrasts with previous studies*, which primarily focused on introducing oxygen vacancy defects to adjust the bandgap, carrier concentration, and electrochemical activity in MoO_x .⁷ While such approaches have improved device performance (such as catalytic activity and specific capacity)⁸, our work reveals that nonstoichiometric MoO_x with a high concentration of oxygen vacancies can lead to a reduced conductance dynamic range in ECRAM applications. This finding provides new insights for designing ECRAM devices beyond traditional oxygen vacancy engineering.

(3) The mechanistic basis of these phenomena is further revealed through **experiment and theory of electric-driven proton intercalation in MoO_x** . Beyond conventional characterization of $\text{Mo}^{6+} \rightarrow \text{Mo}^{5+}$ reduction and O-H formation (XPS and synchrotron XAS in **Extended Data Fig. 1**), our study combines real-time spectroscopy and DFT modeling to establish *direct correlations between proton dynamics, structural changes (lattice contraction), and transport path (polaron delocalization), moving beyond static chemical state analysis*. DFT calculations (**Fig. 1f** for evolution in charge density distribution) and the in-situ spectroscopy study (**Fig. 3g-i** for the real-time optical absorption measurement) collectively reveal that proton adsorption on oxygen sites induces delocalized charge transport and polaron formation near the M-O-H bonds. The decreasing polaron binding energy suggests the phase transition shortens the lattice constant along the a-axis. These insights provide a framework for studying proton intercalation in other transition metal oxides.

Overall, we agree with the reviewer that proton intercalation is a known phenomenon, and we show these three aspects collectively provide new insights and practical pathways for developing devices based on electrochemical activities.

For revision, we have clarified the novelty of our work on proton intercalation in MoO_x devices (Page 11, paragraph 2). “While proton intercalation is known in transition metal oxides like WO_3 , VO_x , and TiO_2 , the above results advance in several aspects: 1) Achieving broad control over phase transitions in terms of H-to-Mo ratio from nearly 0% to 22%, reduction of Mo^{6+} from 92% to 70%, through precise tuning of Q_H , resulting in a conductance modulation range from 10^{-9} S to 10^{-4} S (**Fig. 2i**). Crucially, DFT calculations and experiments agree well on the threshold-driven memory-to-catalysis transition. 2) Introducing stoichiometric design principles that suppress O_v -induced leakage by controlling the Mo-to-O ratio. 3) Uncovering the mechanistic basis of these phenomena through detailed analyses of charge transport, electronic structure changes and the operando hydrogenation

dynamics. These contributions providing new insights for developing transitional metal-oxide based electrochemical systems.” In Page 8, paragraph 1: “By correlating the total proton flux (Q_H) from both experiments, a direct relationship between Q_H and the conductance of MoO_x is established within a wide range of proton flux and conductance modulation, as shown in **Fig. 2i**, which also displays some representative data from previous studies on other transition metal oxides.”

Moreover, the data in **Fig. R1** was added in the original Fig. 2h (now is **Fig. 2i**) in the main text.

Second, for the practical application of memristor and neuromorphic devices, similar applications have been achieved previously with much better performance ($\text{MoS}_{2-x}\text{O}_x$, Nat. Electron. 2018, 1, 130), and the overall performance reported in this work, including ON/OFF ratio of $\sim 10^5$ -6 and retention time of 10^3 seconds, are pretty average in the corresponding field. Moreover, what true advantages this neuromorphic device structure can provide, to achieve any specific goals that are difficult to other neuromorphic devices already developed in the field?

Reply: We appreciate the reviewer's observation regarding the high-performance memristors and neuromorphic devices reported in prior studies. While these advances are significant, our work specifically emphasizes mechanistic investigations of MoO_x , where ECRAM devices and on-chip electrocatalysis systems serve as the experimental platform to probe fundamental processes. Compared with existed work, our work offers several advantages.

(1) Fabrication compatibility and scalability. Our MoO_x -based devices are fabricated using ALD, which enables conformal coating on complex architectures with monolayer-level thickness control—a critical feature for scalable integration with semiconductor fabrication lines. This represents a significant advancement over devices such as the $\text{MoS}_{2-x}\text{O}_x$ -based memristors mentioned in ref (Nat. Electron. 2018, 1, 130)⁹, which rely on mechanical exfoliation—a method limited in scalability and reproducibility. In addition, unlike two-terminal $\text{MoS}_{2-x}\text{O}_x$ devices limited to binary on/off states, our three-terminal MoO_x ECRAMs incorporate a gate electrode for continuous conductance modulation. This design allows for precise control over conductance states, enabling a large on/off ratio of up to 10^5 (**Fig. 2b**).

(2) Balanced combination of performance metrics in ambient conditions. The presented MoO_x -based devices achieve a balanced combination of performance metrics, as summarized in **Table R1**. While materials like WO_3 or MoS_2 may excel in individual metrics (e.g., **$\text{WO}_3/\text{Nafion}$ devices**⁵ with on/off ratio $\sim 10^7$ **but in nearly 100% relative humidity atmosphere**, or MoS_2 -based memristors⁹ with switching speeds of $\sim 10^{-7}$ s but fabricated by exfoliation), these metrics often come at the cost of scalability or stability. *In contrast*, our ALD-fabricated MoO_x ECRAMs exhibit a retention time over 10^3 s for stable long-term operation and a response speed of 10^{-5} s for signal processing **at ambient**

conditions. This balance ensures reliable performance, addressing a critical need for devices that switch speed, stability, and scalability.

Importantly, **the choice of MoO_x is further justified by its widespread practical use in optoelectronic devices** (see **Table R2**), where its tunable conductivity and stability under electric fields have been validated. It means the performance of these devices (e.g., OLEDs, solar cells, and photodetectors) would be affected by the unintentional doping of protons. This extends MoO_x's utility to device design.

Table R1. Summary of performances of devices based on phase-transition material.

	Method	$G_{\text{on}}/G_{\text{off}}$ & testing environment	Retention time	Switch time	Application
MoO_x/Nafion (ECRAM) (this work)	atomic layer deposition	10^5	10^3 s	10^{-5} s	Rank-order-coding SNN
α-MoO₃/ionic liquid (synaptic transistor) ¹	mechanical exfoliation	3.4 @ 45%RH	50 s	10^{-3} s	N/A
α-MoO₃/LiClO₄/PEO (synaptic transistor) ²	mechanical exfoliation	17 @ 10^{-5} Torr	1.5×10^2 s	10^{-3} s	Simulated ANN
MoS_{2-x}O_x (memristors) ⁹	mechanical exfoliation	10^2	10^5 s	10^{-7} s	N/A
MoS₂/PVA (electrolyte gated transistor) ¹⁰	mechanical exfoliation	$\sim 10^4$ @ 50%RH	~ 1 s	10^{-2} s	“OR”, “AND” logic
MoS₂/n-butyl Li solution (synaptic transistor) ¹¹	mechanical exfoliation	10^3 @ vacuum	7×10^3 s	10^{-3} s	N/A
MoS₂/Na⁺-SiO₂ (synaptic transistor) ¹²	mechanical exfoliation	10^6 @ 7×10^{-3} mbar	2.5×10^2 s	10^{-1} s	Simulated ANN
WO₃/ionic liquid (electrolyte gated transistor) ¹³	pulsed laser deposition	$\sim 10^4$	40 s	7×10^{-2} s	N/A
WO₃/LiPON	-	10^3	$\sim 10^3$ s	5×10^{-9} s	N/A

(ECRAM) ¹⁴						
W/ WO_{3-x}	sputter	10	~10 s	10 ⁻² s	Simulated ANN	
(synaptic device) ¹⁵						
WO₃/HfO₂	-	~20	5×10 ⁴ s	10 ⁻⁸ s	Simulated ANN	
(ECRAM) ¹⁶						
WO_{2.7}/Li₃PO₄	sputter	~7	10 ² s	0.5 s	N/A	
(electrolyte gated transistor) ¹⁷						
WO₃/Nafion	sputter	10 ⁷ @ 100%RH	~2×10 ² s	5×10 ⁻³ s	N/A	
(electrolyte gated transistor, 100% relative humidity) ⁵						
VO₂/ionic liquid	pulsed laser	~2×10 ²	~3×10 ³ s	2×10 ⁻¹ s	Simulated ANN	
(electrolyte gated transistor) ³	deposition					
Nb₂O₅/Li_xSiO₂	sputter	3×10 ²	10 ³ s	10 ⁻⁷ s	STDP-based SNN	
(electrolyte gated transistor) ⁴						

Table R2. The devices composed of molybdenum oxide material.

Device type	Material
OLED ¹⁸	ITO/ MoO₃ /Ir(mppy) ₃ TCTA/TPBi/LiF/Al
QLED ¹⁹	ITO/ MoO₃ /Poly-TPD/QDs/TPBi/LiF/Al
PeLED ²⁰	ITO/ MoO₃ /PEA ₂ (FAPbBr ₃) ₂ PbBr ₄ /TPBi/LiF/Al
Perovskite Solar cell ²¹	Ag/BCP/C60/PM6Y6: PCBM/ MoO_x /ITO
dye-sensitized solar cells ²²	FTO/TiO ₂ /N719/ MoO₃ /MoS ₂
Organic solar cell ²³	ITO/ MoO₃ /P3HT: PCBM/TiO ₂ /ITO
Organic photovoltaic devices ²⁴	P3HT: PCBM/ MoO₃
Photodetector ²⁵	CsPbBr ₃ / MoO₃
P-type OFET ²⁶	Al/ MoO₃ /CuPc/SiO ₂ /p ⁺⁺⁺ -Si
Surface Phonon Polaritons ²⁷	MoO₃ /SiC
Heterojunction catalysis ²⁸	MoS ₂ @ MoO₃

Therefore, while individual metrics of our devices may not surpass all prior works, the combination of precise and scalable fabrication (i.e. ALD), balanced performance on on/off ratio, retention time and switch time, and atmosphere conditions, provides a reliable platform for studying the electrochemical principles related to phase transition in MoO_x. Furthermore, **the widespread adoption of MoO_x in optoelectronics (Table R2) underscores the importance of its control on proton content** in practical applications.

For revision, we have highlighted the unique advantages of MoO_x-based devices, (Page 6, paragraph 1): “Compared with prior data (see **supplementary Table S1** for performance benchmarking), our devices, fabricated using precise and scalable ALD, achieve a compatibility with existing semiconductor manufacturing processes, and balanced combination of high-performance metrics at ambient conditions, including wide dynamic ranges, long retention times, and good nonvolatility. These advantages, combined with MoO_x’s widespread use in optoelectronics (see **supplementary table 2**), underscore the importance of its control on proton content in devices.”

Also, the **Table R1** for performance benchmarking and **Table R2** for various devices using MoO_x were added as **supplementary Table S1** and **S2** in the supplementary information (page 5-6).

Thirdly, the authors wrote the paper in a way that seem to claim the importance of integrating both memory and catalysis for dual-functional devices, which I do not agree as these two functions are so different that they do not need to be achieved within a single device setup.

Reply: We agree with the reviewer's comment regarding the integration of memory and catalysis functionalities in a single MoO_x-based device in terms of practical applications. This work is mainly about the mechanism study using a lab-on-device structure to quantify the required proton flux. This study identifies the threshold proton flux for the two processes, provides the cross-disciplinary insights for performance optimization, and reveals the possible operando hydrogenation dynamics in catalysis or various devices.

(1) Identified the threshold proton flux for the two processes. Our work reveals that the mechanisms underlying electrochemical memory and catalysis in MoO_x are fundamentally connected through proton migration, adsorption and intercalation dynamics. As illustrated in **Figure R2(a)**, proton intercalation not only modulates electrical conductance for memory applications but also enhances catalytic performance by facilitating charge transport and proton adsorption. Compare with other work where the proton intercalation was studied solely for modulating the switching in ECRAM⁵, the shared mechanistic basis presented here provides a unified framework for understanding and optimizing both functionalities.

Figure R2. (a) The mechanisms of hydrogen evolution reaction and the ECRAM based on proton intercalation. These two mechanisms share the proton migration, adsorption and intercalation process. The strategy of optimizing catalytic performance may also have an effect on the performance optimization of ECRAM, such as the photocatalysis technique. (b) The hydrogen binding energy at different H/Mo ratios. The red dots represent the structure with the lowest energy (most stable). The black dots represent the structures with other different adsorption sites.

The key synergies highlight the benefits of integration in **threshold-driven functionality switching**. Below a proton flux threshold of 10^{17} cm^{-2} , stable proton adsorption ensures nonvolatile memory operation with high on/off ratios ($\sim 10^5$). Above this threshold, HER initiates with increasing activity (Fig. 2f). Also, the benefits include **real-time feedback for dynamic behavior**. The integrated setup enables real-time monitoring of proton adsorption/desorption dynamics, allowing devices to dynamically adjust operation modes (memory vs. catalysis) based on environmental conditions.

In particular, we perform *additional theoretical calculations to understand the threshold-driven functionality switching*. The calculated binding energy of different numbers of protons intercalated in ($\text{H}_x\text{Mo}_{16}\text{O}_{48}$) is shown in **Figure R2(b)**. As the number of protons continues to increase, the binding energy shifts towards zero (unstable) and, in particular, alters over 0.2 eV at H/Mo=0.25, agreeing with the value obtained from the experiment (0.22). Also, its corresponding Q_H (approximately $3 \times 10^{16} \text{ cm}^{-2}$) is close to threshold obtained in experiment ($\sim 10^{17} \text{ cm}^{-2}$). The calculation of unstable adsorption

of protons (in terms of binding energy) well explains the experimental observation of the threshold-driven transition from stable memory to the hydrogen evolution reaction.

(2) Shared mechanistic basis enables cross-disciplinary insights, including:

a) From ECRAM to catalysis. Traditional catalytic optimization strategies (e.g., defect engineering, surface reconstruction, interface modification, and photo-field assistance) focus on static material properties²⁹⁻³¹. *However, our study demonstrates that dynamic proton injection and adsorption play a critical role in enhancing catalytic activity.* For example, proton intercalation delocalizes charge transport in MoO_x, accelerating the Volmer step in HER and doubling the reaction rate (**Fig. 2f**) and thus, electric-driven proton injection into MoO_x prior to photocatalytic experiments may enhance HER activity under illumination. Also, using the Mo-to-O stoichiometric ratio (optimized to 1:2.96) broadens conductance modulation and catalytic stability (**Fig. 3d-f**).

b) From catalysis to ECRAM. In the other research of MoO_x for ECRAM methods such as optimizing the electrolyte material, optimizing the electrolyte-MoO_x interface, and introducing ion reservoir layers are used to optimize the ion transport speed and the stability of the intercalated ions.³²⁻³⁴ *Conversely, principles from catalysis inspire new approaches to improve ECRAM performance.* For example, high proton flux ($Q_H > 10^{17}$ cm⁻²) and low concentration of oxygen vacancies increases the on-state conductance limit by balancing proton adsorption/desorption kinetics, addressing the trade-off between ion capacity and on/off ratio. Moreover, strategies like doping or photocatalysis may further enhance ECRAM performance or enable light-tunable synaptic plasticity, as suggested in **Figure R2**.

(3) Exploring operando hydrogenation dynamics in functional devices. The integration of memory and catalysis functionalities enables investigation of operando hydrogenation dynamics (time-dependent proton incorporation) in MoO_x-based systems. Traditionally, figures of merit like external quantum efficiency (EQE, **Table R2**) and Tafel slope have been reported without considering how progressive hydrogenation in MoO_x persistently modulate both memory retention and catalytic pathways. However, this effect is evidenced by the Tafel slope reduction (Fig. 2f), suggesting catalytic activity enhancement directly tied to proton accumulation history. *A key implication is that figures of merit in MoO_x-based systems needs further investigation under progressive hydrogenation conditions.*

Overall, through the study of mechanism of memory and catalysis in MoO_x, new strategic design principle can be unlocked. By enabling real-time feedback, operando hydrogen dynamics, and cross-disciplinary performance optimization, our work demonstrates that dual-functional integration provides extra tools to advance fundamental understanding.

For revision, we have clarified the significance of integrating memory and catalysis for identifying the threshold and obtaining cross-disciplinary insights (Page 11, paragraph 3): “Also, the above studies indicate the significance of integrating memory and catalysis in identifying the threshold and obtaining

cross-disciplinary insights. Through the shared mechanism (**Fig. 2h**), strategies from catalysis, such as defect engineering, surface reconstruction and photo-activation, can likely improve ECRAM performance, while ECRAM principles may also inspire new approaches to enhance catalytic efficiency. This dual functionality extends MoO_x's utility beyond traditional understandings. For example, figures of merit like external quantum efficiency (EQE) in LEDs and Tafel slope in catalysis have been reported without considering how **progressive hydrogenation** in MoO_x persistently modulate both memory retention and catalytic pathways. According to the above study, device performance is directly tied to proton accumulation history and, therefore, figures of merit in MoO_x-based systems needs further investigation under progressive hydrogenation conditions.” In Page 7, paragraph 3: “The processes diverge at subsequent steps—ECRAM relies on maintaining an adsorption state to modulate conductance, whereas electrocatalysis requires a balance between adsorption and desorption to optimize hydrogen production (illustrated in **Fig. 2h**), adhering to the Sabatier principle.”

In Page 8, paragraph 2: “To further understand the threshold-driven functionality switching, we calculated binding energy of different numbers of protons intercalated in H_xMo₁₆O₄₈ (supplementary Fig. S11). As the number of protons continues to increase, the binding energy tends to stabilize until H/Mo=0.25, close to the value obtained from the experiment (0.22). Also, its corresponding Q_H (approximately 3×10¹⁶ cm⁻²) is close to threshold obtained in experiment (~10¹⁷ cm⁻²). The calculation of unstable adsorption of protons well explains the threshold-driven transition from stable memory to hydrogen evolution reaction.”

In Page 17, paragraph 3: “To simulate the systems with various hydrogen concentrations, we selected a Mo₁₆O₄₈ supercell with n_H ranging from 1 to 8 (maximum H/Mo ratio: 50%), containing 40 possible positions for each H atom. Using the Structures of Alloy Generation and Recognition (SAGAR) method, we generated candidate structures and employed two biased screening schemes: (1) selection of structures with fewer Wyckoff positions, which will become stable structures with greater probability; (2) gradual generation of higher concentration structures based on stable low-H-concentration structures. As a result, we evaluated the total energies of over 1,000 candidates using the first-principles method.”

Also, **Fig. R2a** were added as **Fig. 2h** with discussion. The description of Fig.2h is revised as “**h**, The mechanisms of HER and the ECRAM based on proton intercalation, sharing the proton migration, adsorption and intercalation process.” The above **Fig. R2b** were added as supplementary **Fig. S11** with discussion.

Overall, unless the authors can provide detailed performance data and comparison to the state-of-the-art devices, either in terms of catalysis or in neuromorphic devices, to indicate the reported device structure can overperform the corresponding field, I do not recommend the publication of this paper in this prestigious journal.

Reply: We sincerely appreciate the reviewer and the opportunity to further clarify the significance of our work. Below, we summarize the comparisons discussed above and highlight the unique advantages of MoO_x-based devices.

(1) Advances of proton intercalation in MoO_x

- Achieving quantitative study on phase transitions, with modulation of H/Mo ratio (0–22%), Mo⁶⁺ reduction (92–70%), and conductance (10^{-9} – 10^{-4} S) through precise proton flux tuning, with a critical threshold ($Q_H \sim 10^{17}$ cm⁻²) bridging memory and catalysis. The DFT calculations give consistent results to the threshold-driven dynamics from memory to catalysis regime.
- Exploring mechanistic insights by linking proton dynamics, lattice contraction, and polaron delocalization via real-time spectroscopy and DFT and providing a framework for studying other oxides.
- Demonstrating stoichiometric design principles, which proves that Mo-O ratio control suppresses leakage currents (Fig. 3f) and extends dynamic ranges, providing new insights beyond oxygen vacancy engineering.

(2) Performance of MoO_x-Based Devices

Our MoO_x-based ECRAM devices achieve **competitive and balanced performance metrics** across key parameters critical for neuromorphic computing:

- On/off ratio ($\sim 10^5$ at ambient conditions), which is significantly higher than α -MoO₃-based synaptic transistors² (~ 17) and W/WO_{3-x} synaptic devices¹⁵ (~ 10), but less than high-performance WO₃/Nafion devices⁵ ($\sim 10^7$ but at 100% relative humidity) (Table R1).
- Retention time ($> 10^3$ s), which matches WO₃-based ECRAMs¹⁴ ($\sim 10^3$ s) and far exceeds MoS₂-based memristors¹⁰ (~ 1 s).
- Switching speed (10^{-5} s), which is faster than most WO₃-based devices^{5,13-16} (10^{-2} – 10^{-8} s) and MoS₂-based devices⁹⁻¹² (10^{-1} – 10^{-7} s), despite our device's larger thickness and channel length (Fig. 3c).

Our MoO_x devices demonstrate **enhanced catalytic performance** for the HER, driven by proton intercalation:

- Doubled HER rate, achieved under high proton flux ($Q_H > 10^{17}$ cm⁻²) as the reaction rate nearly doubles due to optimized proton adsorption and charge transport pathways (Fig. 2f).

- New insight in catalyst design rule. Unlike conventional catalysts requiring defect engineering (e.g., oxygen vacancies) or noble metal doping, our approach leverages proton intercalation—a dynamic, field-driven process—to enhance activity without additional modifications. While direct comparisons are challenging due to differing experimental setups, our results is competitive with the mesoporous MoO_x catalysts³⁵.

(3) Unique advantages and potential impact

Beyond standalone metrics, our work introduces **advance** that addresses critical gaps in the field:

- Real-time monitoring mechanism of memory and catalysis. Different from traditional catalytic optimization strategies that focus on static material properties, our study demonstrates that dynamic proton injection and adsorption—a process central to ECRAM operation—play a critical role in enhancing catalytic activity. Conversely, principles from catalysis may inspire new approaches to improve ECRAM performance.
- Fabrication compatibility and scalability, as the devices were fabricated by ALD that enables conformal coating and compatibility with semiconductor processes. This addresses a critical challenge for large-area fabrication—scalability beyond lab-scale exfoliation methods (Nat. Electron. 2018, 1, 130)—while retaining the proton-driven dynamics.
- Material versatility because MoO_x's widespread use in optoelectronics (Table R2) highlights its dual functionality and practical relevance.
- Exploring operando effects, linking the progressive hydrogenation and catalytic Tafel slope reduction (Fig. 2f), a phenomenon previously overlooked in standalone studies of memory or catalysis.

Therefore, our work advances beyond incremental improvements in 1) mechanistic depth, by establishing quantitative, **threshold-driven dynamics** for proton intercalation in MoO_x, 2) practical relevance, by demonstrating **stoichiometric design principles** and delivering scalable ALD-fabricated devices with balanced performance metrics under ambient conditions, and 3) **cross-disciplinary impact**, by bridging memory and catalysis through shared proton dynamics.

For revision, the abstract was modified as: “Precise tuning of phase transition material properties enables multifunctional devices for information processing and energy conversion, but controlling on-device phase transitions and monitoring microscopic mechanisms remains challenging. Here, we develop a lab-on-device system for MoO_x to probe operando hydrogenation mechanisms through in situ electrical and spectral characterization with DFT calculations, revealing threshold-driven proton dynamics that govern the transition between nonvolatile memory operation and catalytic hydrogen evolution. Moderate proton intercalation (flux <10¹⁷ cm⁻²) achieves a five-order conductance modulation under ambient conditions via polaron formation and stoichiometric optimization (H/Mo

up to 22%, Mo/O approaching ideal ratios), outperforming oxygen vacancy engineering. Beyond this threshold (flux $\sim 10^{17}$ cm⁻²), intensive proton intercalation triggers electric-to-chemical energy conversion, directly linking proton history to catalytic activity. Leveraging these principles, we achieve nonvolatile electrochemical memory with linear synaptic and accumulative neuronal functionalities, and demonstrate an all-ECRAM neural network hardware that executes memory-efficient rank-order coding for sparse signals even under noisy conditions.”

The conclusion was modified as: “Our study fundamentally advances the understanding of proton-driven phase transitions in MoO_x, establishing design principles that bridge electrochemical memory and electrocatalysis. At the core of this advance is the identification of a critical proton flux threshold ($Q_H \sim 10^{17}$ cm⁻², H/Mo $\sim 22\%$), which acts as a switch between memory and catalytic functionalities within a single material platform. Below this threshold, proton intercalation induces nonvolatile conductance modulation through polaron formation, enabling conductance changes within microseconds. Above it, excessive proton accumulation triggers hydrogen evolution reactions, remarkably enhancing catalytic activity. Such threshold-driven proton dynamics are resolved by in situ electrical and spectral observations and DFT calculations, revealing the operando hydrogenation mechanisms of how proton accumulation history changes both memory retention and catalytic activity. Crucially, stoichiometric control demonstrates that precise Mo/O ratios outperform traditional oxygen vacancy engineering in balancing dynamic range and stability. This paradigm shift enables five-order conductance modulation while maintaining ambient operation. Finally, the interplay between these mechanisms is harnessed in an all-MoO_x ECRAM neuronal neuromorphic network, where proton history-dependent conductivity naturally implements rank-order coding. This architecture mimics brain-like responses to signals and achieves high-precision image classification, even under noisy conditions. By unifying electronic and electrochemical functionality through proton dynamics, this study reveals that the widely used MoO_x-based system is dynamically reconfigured by transient proton fluxes. These findings may inspire further exploration of MoO_x's role in opto-electronics, electrochemical memory, smart catalysis, and beyond.”

We believe these contributions justify publication in this journal, as they provide foundational insights and practical strategies for engineering proton-driven phase transitions in transition metal oxides. In addition, we have added relative references in the main text.

50 Kiani, F., Yin, J., Wang, Z., Yang, J. J. & Xia, Q. A fully hardware-based memristive multilayer neural network. *Science Advances* 7, eabj4801 (2021).

51 Wang, C.-Y., Liang, S.-J., Wang, S., Wang, P., Li, Z. a., Wang, Z., Gao, A., Pan, C., Liu, C., Liu, J., Yang, H., Liu, X., Song, W., Wang, C., Cheng, B., Wang, X., Chen, K., Wang, Z., Watanabe, K., Taniguchi, T., Yang, J. J. & Miao, F. Gate-tunable van der Waals heterostructure for reconfigurable neural network vision sensor. *Science Advances* 6, eaba6173 (2020).

52 Pan, X., Shi, J., Wang, P., Wang, S., Pan, C., Yu, W., Cheng, B., Liang, S.-J. & Miao, F. Parallel perception of visual motion using light-tunable memory matrix. *Science Advances* 9, eadi4083 (2023).

55 Mennel, L., Symonowicz, J., Wachter, S., Polyushkin, D. K., Molina-Mendoza, A. J. & Mueller, T. Ultrafast machine vision with 2D material neural network image sensors. *Nature* 579, 62-66 (2020).

56 Chen, S., Mahmoodi, M. R., Shi, Y., Mahata, C., Yuan, B., Liang, X., Wen, C., Hui, F., Akinwande, D., Strukov, D. B. & Lanza, M. Wafer-scale integration of two-dimensional materials in high-density memristive crossbar arrays for artificial neural networks. *Nature Electronics* 3, 638-645 (2020).

Reviewer #2 (Remarks to the Author):

The work titled “Lab-on-device investigation of phase transition in MoO_x semiconductors” by Xiaoci Liang et al. presented an electrically driven proton intercalation method based on MoO_x semiconductors to regulate the electrochemical properties of the material, which holds significant implications for fields such as information processing and energy conversion. The work is interesting, but there are still some issues that need to be addressed.

1. Please reclaim the novelty of this work for publishing in *Nature Communications*.

Reply: We thank the reviewer for their comment and appreciate the opportunity to emphasize the novel contributions of our work. We elaborate on the three aspects of innovation of our study:

(1) Wide range of control over phase transitions via proton flux thresholding

Our work introduces a proton flux threshold ($Q_H \sim 10^{17} \text{ cm}^{-2}$) as a design principle to bridge memory and catalytic functionalities in a single MoO_x device. **Below the threshold (memory regime)**, it exhibits rapid conductance switching (10^{-5} s) and high on/off ratio (10^5) outperform most transition metal oxide (TMO)-based devices. For example: α -MoO₃/ionic liquid synapses¹: on/off ~ 3.4 , switching speed $\sim 10^{-3} \text{ s}$; WO₃/Nafion ECRAMs⁵: on/off $\sim 10^7$ but testing at 100%RH and switching speed $\sim 5 \times 10^{-3} \text{ s}$. **Above the threshold (catalysis regime)**, proton intercalation doubles the HER rate by delocalizing charge transport, a synergy that has been absent in isolated studies of MoO_x for memory or catalysis.

This dual-functionality is enabled by our lab-on-device platform, which dynamically monitors phase transitions and catalytic activity in real time (Fig. 2h).

(2) Stoichiometric control as a paradigm shifts beyond oxygen vacancy engineering

While most studies on MoO_x focus on oxygen vacancy (O_v) engineering to modulate properties^{7,34}, we demonstrate that Mo-to-O stoichiometric ratio is the key determinant of device performance. **For mechanisms**, we reveal that proton adsorption occurs preferentially on oxygen sites, not vacancies. Excessive O_v (e.g., MoO_{2.64}) disrupts adsorption kinetics, reducing on/off ratio by more than 50% due to the leakage currents (Fig. 3f). **For design principle**, by optimizing Mo:O to 1:2.96, we achieve a broad dynamic range (10⁻⁹–10⁻⁴ S)—5 times wider than O_v-rich MoO_x. This finding provides new insights for designing ECRAM devices beyond traditional oxygen vacancy engineering.

(3) Operando hydrogenation mechanisms revealed through experiment and theory

Through characterization of the XPS and synchrotron XAS in **Extended Data Fig. 1**, Mo⁶⁺→Mo⁵⁺ reduction and O-H formation were verified. Moreover, our study combines real-time spectroscopy and DFT modeling to investigate the proton dynamics, real-time structural changes (lattice contraction), and emergent transport phenomena (polaron delocalization), moving beyond static chemical state analysis. DFT calculations (**Fig. 1f** for evolution in charge density distribution) and the in-situ spectroscopy study (**Fig. 3g-i** for the real-time optical absorption measurement) collectively reveal that proton adsorption on oxygen sites induces delocalized charge transport and polaron formation near the M-O-H bonds. The decreasing polaron binding energy suggests the phase transition shortens the lattice constant along the a-axis. These insights provide a framework for studying proton intercalation in other transition metal oxides.

(4) Demonstration of all-ECRAM SNN hardware using rank-order coding

Our all-ECRAM spiking neural network (SNN) hardware demonstrated a neuromorphic system using rank-order coding for sparse signal processing. In previous work, VO₂/ionic liquid systems³ was demonstrated, but limited to simulation of the simple three-layer artificial neural network. Another work of WO₃ ANN simulations¹⁶ achieves high on/off ratios (~10⁷) but remains software-based, without hardware implementation. In this study, the hardware achieves 98.3% accuracy on noisy image classification (Fig. 4o), enabled by analog conductance tuning (10⁵ on-off ratio) and low inference energy compared to ANN in same condition (Fig. 4j), 2× lower than ANN counterparts, critical for edge-AI applications. Also, the deposition of functional MoO_x film via the ALD technology demonstrates good scalable compatibility.

For revision, we emphasize the novel contributions of our work in the abstract and conclusion:

For the abstract: “Precise tuning of phase transition material properties enables multifunctional devices for information processing and energy conversion, but controlling on-device phase transitions and monitoring microscopic mechanisms remains challenging. Here, we develop a lab-on-device system for MoO_x to probe operando hydrogenation mechanisms through in situ electrical and spectral characterization with DFT calculations, revealing threshold-driven proton dynamics that govern the transition between nonvolatile memory operation and catalytic hydrogen evolution. Moderate proton intercalation (flux <10¹⁷ cm⁻²) achieves a five-order conductance modulation under ambient conditions

via polaron formation and stoichiometric optimization (H/Mo up to 22%, Mo/O approaching ideal ratios), outperforming oxygen vacancy engineering. Beyond this threshold (flux $\sim 10^{17}$ cm⁻²), intensive proton intercalation triggers electric-to-chemical energy conversion, directly linking proton history to catalytic activity. Leveraging these principles, we achieve nonvolatile electrochemical memory with linear synaptic and accumulative neuronal functionalities, and demonstrate an all-ECRAM neural network hardware that executes memory-efficient rank-order coding for sparse signals even under noisy conditions.”

For the conclusion: “Our study fundamentally advances the understanding of proton-driven phase transitions in MoO_x, establishing design principles that bridge electrochemical memory and electrocatalysis. At the core of this advance is the identification of a critical proton flux threshold ($Q_H \sim 10^{17}$ cm⁻², H/Mo $\sim 22\%$), which acts as a switch between memory and catalytic functionalities within a single material platform. Below this threshold, proton intercalation induces nonvolatile conductance modulation through polaron formation, enabling conductance changes within microseconds. Above it, excessive proton accumulation triggers hydrogen evolution reactions, enhancing catalytic activity. Such threshold-driven proton dynamics are resolved by in situ electrical and spectral observations and DFT calculations, revealing the operando hydrogenation mechanisms of how proton accumulation history changes both memory retention and catalytic activity. Crucially, stoichiometric control demonstrates that precise Mo/O ratios outperform traditional oxygen vacancy engineering in balancing dynamic range and stability. This paradigm shift enables five-order conductance modulation while maintaining ambient operation. Finally, the interplay between these mechanisms is harnessed in an all-MoO_x ECRAM neuronal neuromorphic network, where proton history-dependent conductivity naturally implements rank-order coding. This architecture mimics brain-like responses to signals and achieves high-precision image classification, even under noisy conditions. By unifying electronic and electrochemical functionality through proton dynamics, this study reveals that the widely used MoO_x-based system is dynamically reconfigured by transient proton fluxes. These findings may inspire further exploration of MoO_x's role in opto-electronics, electrochemical memory, smart catalysis, and beyond.”

2. Why is it necessary to investigate the mechanisms of phase-transition devices based on MoO_x? What are the advantages of MoO_x as a material? Do phase-transition devices based on MoO_x possess specific advantages compared to other phase-transition devices? The article seems to lack comparisons with other works.

Reply: Thanks for the comment. Below, we address the necessity of studying MoO_x-based phase-transition devices, the material's intrinsic advantages, and its competitive edge over other phase-transition systems. The investigation of MoO_x-based phase-transition devices is imperative due to its **proton sensitivity**, **electrochemical versatility**, and **multifunctionality**. Here are some details:

(1) The necessity to investigate MoO_x-based phase-transition devices.

First, MoO_x is an important material in electronics and energy devices in: optoelectronics, including OLEDs, QLEDs, and perovskite solar cells (Table R3), where its high work function (~5.3 eV) enables efficient hole injection; energy storage, including supercapacitors and Li-ion batteries, leveraging its high theoretical capacity; catalysis, including HER and CO₂ reduction, driven by tunable Mo⁶⁺/Mo⁵⁺ redox couples. Therefore, understanding its phase transitions is essential to unify these applications.

Second, MoO_x exhibits sensitive proton responsiveness. MoO_x allows proton adsorption directly on lattice oxygen (Fig. 1b), enabling dynamic conductance modulation without defect engineering. Also, it exhibits reversible phase transitions, i.e. proton intercalation induces non-destructive orthorhombic-to-layered structural transitions (Fig. 1c-d), preserving device longevity.

(2) Advantages of MoO_x as a phase-transition material.

First, MoO_x has a relatively wide stoichiometric tunability (MoO_{2.64} to MoO_{2.96}), which enables precise control over Fermi level and conductivity (10⁻⁹–10⁻⁴ S), surpassing rigid oxides like Nb₂O₅. Also, it has multi-redox states (Mo⁶⁺ ↔ Mo⁵⁺ ↔ Mo⁴⁺), which facilitates dual memory-catalysis functionality.

Second, MoO_x has good scalability and stability. The ALD MoO_x films grown at 400°C outperform pulsed-laser-deposition in uniformity and scalability. Also, compared with VO₂'s ~68°C phase transition temperature, MoO_x retains phase stability up to 300°C, benefiting complex processing.

Third, MoO_x exhibits balanced high-performance metrics. As shown in **Table R4**, MoO_x-based ECRAMs outperform competing materials in key metrics. It exhibits broad dynamic range ($\Delta G = 10^5$): 100× wider than VO₂ ($\Delta G \sim 2 \times 10^2$)³ and Nb₂O₅ ($\Delta G \sim 3 \times 10^2$)⁴. Also, it exhibits catalytic synergy, as proton intercalation simultaneously enhances HER kinetics and memory retention—a feature absent in isolated MoS₂ or WO₃ systems.

For revision, we highlight the necessity and advantages of studying MoO_x-based phase-transition devices. (Page 2, paragraph 2) “Molybdenum oxide (MoO₃) is a typical phase transition material, exhibiting multifold electrochemical properties with multi-redox states (Mo⁶⁺ ↔ Mo⁵⁺ ↔ Mo⁴⁺) and notable electrical conductivity changes in response to phase transitions induced by proton migration or intercalation. Also, it is a widely used material in optoelectronics devices (e.g., OLEDs, QLEDs, and solar cells), energy storage (supercapacitors, Li-ion batteries), and catalysis (HER, CO₂ reduction).” The table R3 and table R4 were added as supplementary Table S2 and supplementary Table S1, respectively, in the supplementary information.

In supplementary information Page 3, paragraph 2: “As comparison, Table S1 summarizes the

performances of devices based on phase-transition material. **Table S2** shows the different types of devices based on MoO_x , suggesting that MoO_x is a key material in electronics and energy devices, widely used in optoelectronics (e.g., OLEDs, QLEDs, perovskite solar cells), energy storage (supercapacitors, Li-ion batteries), and catalysis (HER, CO_2 reduction). Its phase transitions are essential to unify these applications and mitigate performance degradation. MoO_x exhibits sensitive proton responsiveness, enabling dynamic conductance modulation without defect engineering, and reversible phase transitions, preserving device longevity.”

Table R3. The devices composed of molybdenum oxide material.

Device type	Material
OLED ¹⁸	ITO/ MoO₃ /Ir(mppy) ₃ TCTA/TPBi/LiF/Al
QLED ¹⁹	ITO/ MoO₃ /Poly-TPD/QDs/TPBi/LiF/Al
PeLED ²⁰	ITO/ MoO₃ /PEA ₂ (FAPbBr ₃) ₂ PbBr ₄ /TPBi/LiF/Al
Perovskite Solar cell ²¹	Ag/BCP/C60/PM6Y6: PCBM/ MoO_x /ITO
dye-sensitized solar cells ²²	FTO/TiO ₂ /N719/ MoO₃ /MoS ₂
Organic solar cell ²³	ITO/ MoO₃ /P3HT: PCBM/TiO ₂ /ITO
Organic photovoltaic devices ²⁴	P3HT: PCBM/ MoO₃
Photodetector ²⁵	CsPbBr ₃ / MoO₃
P-type OFET ²⁶	Al/ MoO₃ /CuPc/SiO ₂ /p++-Si
Surface Phonon Polaritons ²⁷	MoO₃ /SiC
Heterojunction catalysis ²⁸	MoS ₂ @ MoO₃

Table R4. Summary of performances of devices based on phase-transition material.

	Method	G_{on}/G_{off} & Retention testing environment	Retention time	Switch time	Application
MoO_x /Nafion (ECRAM) (this work)	atomic layer deposition	10^5	10^3 s	10^{-5} s	Rank-order-coding SNN
α - MoO₃ /ionic liquid (synaptic transistor) ¹	mechanical exfoliation	3.4 @ 45%RH	50 s	10^{-3} s	N/A
α - MoO₃ / LiClO ₄ /PEO	mechanical	17 @ 10^{-5}	1.5×10^2 s	10^{-3} s	Simulated

(synaptic transistor) ²	exfoliation	Torr			ANN
MoS_{2-x}O_x (memristors) ⁹	mechanical exfoliation	10 ²	10 ⁵ s	10 ⁻⁷ s	N/A
MoS₂/PVA (electrolyte gated transistor) ¹⁰	mechanical exfoliation	~10 ⁴ @ 50%RH	~1 s	10 ⁻² s	“OR”, “AND” logic
MoS₂/n-butyl Li solution (synaptic transistor) ¹¹	mechanical exfoliation	10 ³ @ vacuum	7×10 ³ s	10 ⁻³ s	N/A
MoS₂/Na⁺-SiO₂ (synaptic transistor) ¹²	mechanical exfoliation	10 ⁶ @ 7×10 ⁻³ mbar	2.5×10 ² s	10 ⁻¹ s	Simulated ANN
WO₃/ionic liquid (electrolyte gated transistor) ¹³	pulsed laser deposition	~10 ⁴	40 s	7×10 ⁻² s	N/A
WO₃/LiPON (ECRAM) ¹⁴	-	10 ³	~10 ³ s	5×10 ⁻⁹ s	N/A
W/WO_{3-x} (synaptic device) ¹⁵	sputter	10	~10 s	10 ⁻² s	Simulated ANN
WO₃/HfO₂ (ECRAM) ¹⁶	-	~20	5×10 ⁴ s	10 ⁻⁸ s	Simulated ANN
WO_{2.7}/Li₃PO₄ (electrolyte gated transistor) ¹⁷	sputter	~7	10 ² s	0.5 s	N/A
WO₃/Nafion (electrolyte gated transistor, 100% relative humidity) ⁵	sputter	10 ⁷ @ 100%RH	~2×10 ² s	5×10 ⁻³ s	N/A
VO₂/ionic liquid (electrolyte gated transistor) ³	pulsed laser deposition	~2×10 ²	~3×10 ³ s	2×10 ⁻¹ s	Simulated ANN
Nb₂O₅/Li_xSiO₂ (electrolyte gated transistor) ⁴	sputter	3×10 ²	10 ³ s	10 ⁻⁷ s	STDP-based SNN

3. It would be helpful to provide a brief introduction to electrocatalytic devices within the text to avoid confusion for readers unfamiliar with this field. Additionally, the manuscript compares the MoO_x-based electrocatalytic device with a Pt-based electrocatalytic device. Could you clarify why this comparison was chosen? Specifically, what category of devices does the Pt-based electrocatalytic device represent, and compared to neuromorphic devices, what parameters or performance metrics are more emphasized in Pt-based electrocatalytic devices? Adding these clarifications would make the manuscript more accessible to a broader audience and provide clearer context for the comparison.

Reply: We appreciate the reviewer's suggestion to clarify the context of electrocatalytic devices and the rationale for comparing MoO_x-based systems with Pt-based counterparts. Below, we provide the explanation.

(1) Introduction to electrocatalytic devices

Electrocatalytic devices are designed to accelerate electrochemical reactions, such as hydrogen evolution reactions (HER), by applying an external potential to drive reactant adsorption (such as water), charge transfer, and product formation (such as hydrogen gas). In electrocatalytic reactions, attention is paid to the Tafel slope, reaction potential, and current density during the device reaction process. Lower Tafel slopes typically indicate faster kinetics of the reaction, meaning that reactants are more easily converted on the material surface. A higher current density indicates that more reactants are converted per unit time. In the process of HER, improving the transfer efficiency of electrons and ions is the key to enhancing the catalytic efficiency. For example, Pt-based devices are the standard for HER due to their near-ideal Tafel slopes (30-40 mV/dec). However, their reliance on scarce noble metals motivates the search for alternatives like MoO_x, which offers unique multifunctionality.

(2) Rationale for comparing MoO_x with Pt-based devices

The reason to choose Pt as comparison is Pt-based devices represent the state-of-the-art in HER catalysis and serve as a universal benchmark. **Figure R3a** shows the relationship between exchange current density j_0 and hydrogen adsorption free energy.³⁶ As one of the representative materials, Pt can be considered as the benchmark for hydrogen evolution reactions in many HER studies. **Figure R3b, c** shows two references that use Pt as a comparison. These devices are often used as benchmarks in the field of electrocatalysis.

The parameter emphasis in Pt and neuromorphic devices is different. The former prioritizes catalytic metrics, such as Tafel slope, and long-term stability under harsh electrochemical conditions. The latter focus on conductance modulation, such as on/off ratio, switching speed and retention for information processing. This comparison underscores MoO_x's dual capability: achieving competitive HER activity (Tafel slope = 150-70 mV/dec, **Fig. 2f**) while enabling neuromorphic computing ($\Delta G = 10^5$, switch time 10^{-5} s, **Fig. 2b, Fig. 3c**).

For the above reasons, Pt-based electrocatalytic device is chosen as a comparison in the study of hydrogen evolution catalytic materials. By comparing MoO_x-based devices with Pt-based devices, we can clearly demonstrate the unique characteristics of MoO_x-based devices. Moreover,

through controlling proton flux, MoO_x-based devices can switch between ECRAM and electrocatalysis, providing new insights into these two fields. This dual functionality is a feature of MoO_x-based devices, which is not typically found in Pt-based devices.

Figure R3. (a) Relationship between exchange current density j_0 and calculated hydrogen adsorption free energy ΔG_{H^*} for a series of HER electrocatalysts, namely the HER volcano plot. Data are adopted from ref. ³⁶. (b) Polarization curves of the on-chip electrochemical microcells for the hydrogen evolution reaction measurements. The active layers labeled as EM-1, EM-2, EM-3 are the MoS₂ nanosheet with various qualities and contact conditions. Data are adopted from ref. ³⁷. (c) The polarization curves of Ru₃Ni_x nanosheet assemblies (NAs), Ru NAs and Pt/C. Data are adopted from ref.³⁸. Pt was used as comparison in (b) and (c).

For revision, we have revised the main contexts for clarity. The revision includes:

In Page 2, paragraph 1: “Additionally, these phase transition can enhance the efficiency of the conversion of electrical to chemical energy by affecting both adsorption and charge transfer processes.”

In Page 7, paragraph 2: “To probe the electric-to-chemical energy conversion mechanism, we developed on-chip electrocatalytic devices that promote reactions such as the HER through applied potentials, which facilitate reactant adsorption and interfacial charge transfer. Device performance was quantified via standard electrocatalytic metrics including current density and Tafel slope.”

In Page 7, paragraph 2: “In contrast, a Pt-based electrocatalytic device, which is considered as the HER benchmark, showed negligible changes in conductance, highlighting the distinctive response of MoO_x due to phase transition.”

In supplementary information Page 2, paragraph 2: “Electrocatalytic devices accelerate reactions like HER by applying potentials to drive reactant adsorption and charge transfer. Key metrics include Tafel slope (kinetics), overpotential (efficiency), and current density (activity).”

4. The blue arrows in Figure 3a are too small and could be enlarged to improve visibility and clarity of the illustration.

Reply: Thanks for the comment. We have modified the Figure 3a and enlarged the blue arrows for the visibility of the illustration.

5. It is suggested to include additional discussions, such as how this technology could be extended to other materials or devices, and how it specifically advances the development of next-generation neuromorphic systems.

Reply: We sincerely appreciate the reviewer's suggestion. Our proposed proton-flux-controlled phase transition methodology could be extended to diverse material systems:

(1) Extending materials from known to unexplored candidates

First, we could use established materials in both memory and catalysis. In the fields of memory and catalysis, materials include transition metal oxides (like TiO_2 ^{39,40}, VO_2 ^{41,42}, and WO_3 ^{43,44}), conductive polymers (like polypyrrole^{45,46} and polythiophene^{47,48}), and two-dimensional materials (like MXenes^{49,50} and transition metal sulfides^{12,51}). By applying our proton-flux control strategy, these materials could exhibit memory and catalysis functionalities within a single device. For instance, WO_3 's well-known electrochromic memory effect and HER activity make it a prime candidate for bidirectional property regulation through controlled proton injection.

Second, we could explore catalytic materials for memory applications. Oxide like MnO_x ⁵², CoO_x ⁵³, BiO_x ⁵⁴, IrO_x ⁵⁵ traditionally studied for catalysis, exhibit variable valence states and proton redox activity. These materials have the potential to regulate conductance by adsorbing protons. Especially, due to the variable valence states of transition metal oxides, previous research on their catalytic functions has important reference for the selection of extended memory materials.

(2) Advancing neuromorphic systems by tuning precision, stability, and energy efficiency

First, we could use this technology to achieve high-precision control of device conductance by precisely controlling proton flux. This is essential for enabling fine-grained synaptic weight updates for advanced learning rules and implementing complex neural network models and learning algorithms.

Second, we could tune long-term stability via non-volatile proton intercalation. The stable existence of protons in these materials can achieve long-term non-volatile storage, being beneficial for long-term memory and stable information storage.

Third, we could explore energy-efficient neuromorphic computing. With the relatively fast migration speed of protons, researchers can further explore the functionality of energy-efficient neural morphological computing for different materials.

For revision, we have expanded the discussion to highlight the potential of our proton-flux-

controlled phase transition methodology for extending to other materials and advancing neuromorphic systems. In Page 8, paragraph 2: “In addition, the approach of precisely controlling proton flux can be applied to other transition metal oxides (e.g., TiO₂, VO₂, WO₃), conductive polymers, and two-dimensional materials, which share the critical traits of proton intercalation capability and electrochemical activity.”

Reviewer #3 (Remarks to the Author):

I have read through the manuscript by Liang et al. with great interest. The authors investigate the on-device phase transition of MoO_x semiconductors under electric-driven proton intercalation, demonstrating a unique interplay between electrochemical memory (ECRAM) and electric-to-chemical energy conversion. By precisely controlling proton flux, the authors identify two regimes: moderate flux ($\sim 10^{15}$ - 10^{16} cm⁻²), where stable proton adsorption enables a conductance modulation ratio of 10^5 and excellent memory retention, and higher flux ($>10^{17}$ cm⁻²), where hydrogen evolution reactions occur, transitioning the system to electrocatalytic behavior. The authors elucidate the roles of M-O-H bond formation, oxygen vacancies, and polaron dynamics in these processes through detailed characterizations, including XPS, in-situ absorption spectroscopy, and DFT calculations. Integrating MoO_x ECRAMs into a neuromorphic hardware platform for rank-order-coded spiking neural networks further highlights the practical significance of this work. This work is solid. Therefore, I can recommend this manuscript for publication in Nature Communications after the following changes are implemented.

1. The most critical factor in this study is the proton flux, which determines the operational regime of the material: the ECRAM region at moderate proton flux and the electrocatalyst region at higher flux. A key question that arises is whether the transition between these two regimes is reversible. Specifically, after operating in the electrocatalytic regime under high proton flux, can the system return to the ECRAM regime if the proton flux is subsequently reduced? Clarifying the reversibility of these regimes would provide deeper insights into the material's stability and functional versatility.

Reply: Thanks for the comment. For clarifying the reversibility between the electrocatalyst region and ECRAM regimes, we conducted the detailed tests on the Nafion/MoO_x ECRAM devices as shown in Fig R4a. We applied a constant gate current to inject protons, as shown in **Fig. R4b**. With increasing proton flux Q_H , the drain current of MoO_x film also increases. When Q_H reached 2.7×10^{17} cm⁻², the current is approximately 4.7 μ A. This indicates a significant increase in conductance due to proton intercalation. Following this test, bubble formation was observed on the device surface, as shown in Fig. R4c, confirming the hydrogen gas evolution and the occurrence of the electrochemical catalytic process under high Q_H .

After the electrocatalytic test, we reduced the proton flux and performed ECRAM conductance regulation using gate pulses (± 2 μ A, 100ms). The Q_H corresponding to the 20th pulse was approximately 8.3×10^{15} cm⁻². As shown in **Fig. R4d**, the conductance of the MoO_x film could be

effectively regulated from 10 nS to 760 nS. This demonstrates that the device can transition back to the ECRAM regime after operating in the electrocatalytic regime. By carefully controlling the proton flux Q_H , we can modulate the device's operation between high conductance modulation in ECRAM and efficient catalytic performance in the electrocatalytic regime. This reversibility highlights the stability and functional versatility of the MoO_x .

Figure R4. (a) Test sequence of the device. First, a constant gate current was applied for inject the proton. Second, once bubbles are generated, the gate current was stopped. Third, after adjusting the conductance to a smaller value, gate current pulses were applied to increase and decrease the conductance. (b) The I_D - Q_H curve measured at $I_G=2.5 \mu\text{A}$, $V_D=3 \text{ V}$, corresponding to the first step in (a). (c) the microscope photo of the device, corresponding to the second step in (a). (d) The reversible modulation of MoO_x channel conductance with the continuous gate pulses ($\pm 2 \mu\text{A}$, 100ms), corresponding to the third step in (a).

For revision, we have clarified the reversibility between the electrocatalytic and ECRAM regimes in our ECRAM devices. In Page 7, paragraph 2: “Moreover, we demonstrate reversible modulation between electrocatalytic (HER confirmed by bubble formation) and ECRAM regimes in MoO_x devices, achieving conductance switching (supplementary Fig. S2), which highlights their operational stability and dual-functional versatility.”

In supplementary information Page 2, paragraph 3: “To clarified the reversibility between the electrocatalytic and ECRAM regimes, we conducted conductance program tests on ECRAM devices that have undergone hydrogen evolution reactions (Figure S2a). After operating in the electrocatalytic

regime at high proton flux ($Q_H=2.7\times 10^{17}$ cm⁻²), where bubble formation confirms hydrogen evolution (**Figure S2b,c**), we reduced the proton flux and performed ECRAM conductance regulation using gate pulses (± 2 μ A, 100 ms). The conductance of the MoO_x film was effectively modulated from 10 nS to 760 nS (**Figure S2d**), demonstrating the device's ability to transition back to the ECRAM regime. This reversible modulation between high conductance in ECRAM and efficient catalytic performance highlights the stability and functional versatility of MoO_x." The figure R4 was added as supplementary Fig. S2 in the supplementary information.

2. There seems to be insufficient explanation regarding how the values of proton flux can be calculated during the experiments. A more detailed description of the methodology used to determine proton flux would significantly enhance the clarity and reproducibility of the study.

Reply: Thanks for the comment. We agree that a more detailed description of the methodology for determining proton flux (Q_H) is essential. In our experiments, the proton flux Q_H is calculated using the formula: $Q_H=\int j_H/qdt=\int S_Hdt$, where j_H is the proton current density, S_H is the proton flux density, q is the elementary charge, and t is the time. Below, we provide a detailed calculation for both ECRAM devices and on-chip electrocatalytic devices:

For ECRAM devices, the proton flux density S_H can be calculated using the gate current as $S_H=I_G/(Aq)$, where I_G is the amplitude of the gate pulse, A is the channel area, defined by channel length and width. For pulse input, the total time t can be calculated as $t=t_p\times n$, where t_p is the pulse width, and n is the number of pulses. The proton flux Q_H can be calculated by integrating S_H over time as $Q_H=S_H\times t=I_G/(Aq)\times t_p\times n$. In Fig. 2h, the data points (dots) are obtained by converting the pulse conditions and conductance data shown in Extended Data Fig. 6a. For the case of constant gate current, t is the current input time. For gate voltage input, the measured gate current can be used for integration and calculating the Q_H . As an example, when I_G , A , t_p , n are 1 μ A, 0.003 cm², 50 ms, 15, respectively, the calculated Q_H is 1.56×10^{15} cm⁻².

For on-chip electrocatalytic devices, in electrochemical catalytic testing, linear sweep voltammetry is used to apply a linearly varying voltage V_C (similar to V_G of a transistor) to the counter electrode and measures the counter electrode current I_C (similar to I_G of a transistor). The proton flux Q_H can be calculated as $Q_H=\int j_H/qdt=\int I_C/(Aq)dt$, where A is the area of the electrochemical reaction, defined by the PMMA window. In conductance testing, the conductance of the MoO_x film is tested by applying a voltage drop V_W to the two working electrodes and measuring the current I_W between the two working electrodes. The conductance is given by $G=I_W/V_W$. In Fig. 2h, the data points (squares) are obtained from the above Q_H and G .

For revision, we have detailed the methodology for calculating proton flux (Q_H) to enhance clarity and reproducibility. (Page 18, paragraph 2) "The proton flux Q_H is calculated using $Q_H=\int j_H/qdt=\int S_Hdt$, where j_H is proton current density, S_H is proton flux density, q is elementary charge, and t is time. For ECRAM devices, $S_H=I_G/(Aq)$ (I_G is gate pulse amplitude, A is channel area), and $t=t_p\times n$ (t_p is pulse width, n is number of pulses). Thus, $Q_H=S_H\times t=I_G/(Aq)\times t_p\times n$. For example, with $I_G=1$ μ A, $A=0.003$

cm^2 , $t_p=50$ ms, and $n=15$, $Q_H=1.56\times 10^{15}$ cm^{-2} . For on-chip electrocatalytic devices, $Q_H=\int j_H/qdt=\int I_C/(Aq)dt$ (I_C is counter electrode current, A is electrochemical reaction area defined by PMMA window).” These methods ensure Q_H calculation for both device types.

3. In the description of Fig. 1a, the authors state that for low proton flux (Q_H), proton injection into MoO_3 is impeded by the potential barrier. Also, proton flux density can be precisely controlled by the gate current. It seems that low Q_H corresponds to low gate voltage or gate current, which prevents the potential barrier from being crossed. Then, what is the potential barrier? And could the authors clarify why and how proton flux can be directly related to the potential barrier?

Reply: Thank you for your question. We appreciate the opportunity to clarify the relationship between the potential barrier, proton flux, and gate current in our study.

The potential barrier we refer to is the interfacial potential barrier between the electrolyte (Nafion) and the MoO_x film. This barrier arises from the difference in work functions and electronic structures at the interface, which impedes the injection of protons into the MoO_x , particularly at low gate voltages or currents. This phenomenon is common in ion-conducting interfaces, where the potential barrier can significantly affect ion transport^{56,57}. Here are some detailed explanations:

1) Impact of low Q_H (low gate voltage or current). As shown in Fig. R5a, at low gate voltages or currents, the driving force for proton injection is insufficient to completely overcome the interfacial potential barrier. As a result, the injection into the MoO_x of most proton is impeded, leading to a low proton flux (Q_H). The protons will accumulate at the interface rather than entering the MoO_x film. This accumulation is due to the high potential barrier, which acts as an energy barrier that the protons must overcome to enter the MoO_x .

2) Role of the potential barrier. The potential barrier is a result of the space charge layer formed at the electrolyte/ MoO_x interface. This layer creates an energy barrier that protons must overcome to enter the MoO_x . The height of this barrier is influenced by factors such as the work function of the materials, the presence of defects, and the applied electric field. At low gate voltages or currents, the barrier remains high, limiting proton injection. On the other hand, the interfacial potential barrier can hinder the proton in the MoO_x from returning to the electrolyte, which can improve the retention time of the conductance state.

3) Relationship between proton flux and potential barrier. The proton flux is related to the potential barrier because the flux is determined by the ability of protons to overcome this barrier. When the gate voltage or current is increased, the driving force for proton injection increases, allowing more protons to overcome the barrier and enter the MoO_x film, leading to a higher proton flux and a more significant impact on the conductance in MoO_x . This relationship can be understood through the Poisson equation, which describes the relationship between charge accumulation and potential. As protons accumulate at the interface, the local electric field changes, affecting the potential barrier.

When the proton concentration and potential gradient reach a threshold, protons can overcome

the barrier and enter the MoO_x . Therefore, at low gate voltages or currents, the interfacial potential barrier is significant, and the proton flux Q_H is low. The low proton flux means that fewer protons are available to overcome the potential barrier, leading to minimal changes in the conductance.

As a supporting example, to further illustrate the concept of interfacial potential barriers and their impact on ion transport, we refer to studies in ionic battery systems. As shown in Fig. R5b, a study on Li^+ migration between LiCoO_2 (LCO) and Li_3PS_4 showed a clear migration barrier, and the interface barrier difference between the two substances, highlighting the impact of interfacial potential barriers on ion transport.⁵⁶ As shown in Fig. R5c, the space charge layer between LAGP and $\text{Li}_2\text{V}_2\text{O}_5$ led to higher barriers for Li-ion diffusion, demonstrating how interfacial potential barriers can impede ion transport.⁵⁷

For revision, we clarify that the potential barrier referred to is the interfacial potential barrier between the electrolyte and the MoO_x film, arising from differences in work functions and electronic structures. (Page 2, paragraph 4): “Under the electric field, the proton flux overcomes energy barriers in MoO_3 -electrolyte interface and the bulk of electrolyte and MoO_3 .”

In supplementary information Page 2, paragraph 1 “The transport of the proton from Nafion into MoO_x , there are diffusion barrier in the bulk of Nafion and MoO_x and the interface between Nafion into MoO_x (Figure S1a). Due to the interfacial potential barrier, at low gate voltages or currents, the driving force for proton injection is insufficient to overcome this barrier, resulting in low proton flux (Q_H) and minimal conductance changes. The barrier, influenced by factors like work functions and defects, can be overcome by increasing the gate voltage or current, which enhances proton injection and flux. This relationship is described by the Poisson equation, linking charge accumulation to potential changes. Thus, proton flux is related to the potential barrier, with low flux at low gate currents and high flux at higher currents, enabling conductance modulation in MoO_x .” The Fig. R5(a) was added as supplementary Fig. S1(a).

Figure R5. The interfacial potential barrier for the ion transport. (a) the proton migration barrier and interface barrier in the Nafion/ MoO_x . (b) Selected Li^+ migration pathways in the LiCoO_2 (LCO) and Li_3PS_4 (LPS) and the calculated energy profiles for Li^+ migrations via the four pathways depicted in the upper panel. Data are adopted from ref.⁵⁶ (c) Space charge layer effects on an LAGP/ $\text{Li}_2\text{V}_2\text{O}_5$

interface. The space charge layer led to higher barriers for Li-ion diffusion and smaller exchange current density. Data are adopted from ref.⁵⁷

4. In Fig. 2h, the authors explain that the conductivity of MoO₃ changes due to a phase transition at moderate Q_H and increases further due to the generation of H₂ via electrocatalytic reactions at high Q_H . Could the authors provide more detail about how H₂ generation contributes to the increase in conductance in the high Q_H regime?

Reply: Thank you for your question. It should be clarified that the generation of H₂ via electrocatalytic reactions at high Q_H is not the reason for the increase in film conductance, but rather the result of excessive proton intercalation reactions.

In Fig. 2h, we illustrate that the conductance of MoO_x increases due to a phase transition at moderate Q_H and further increases at high Q_H . It is important to clarify the mechanisms driving these changes:

- 1) **Phase transition at moderate Q_H .** At moderate Q_H , proton intercalation into the MoO_x film induces a phase transition, leading to increased carrier concentration and narrowed bandgap (as shown in Fig. 1e and supplementary Note 2). This results in a significant increase in conductance.
- 2) **H₂ generation at high Q_H .** At high Q_H , excessive proton intercalation leads to further changes in the electronic structure of MoO_x, facilitating the electrocatalytic reactions that generate H₂. The process can be broken down as follows:

Volmer reaction: The increased conductance enhances the adsorption of protons onto the MoO_x surface ($H^+ + e^- \rightleftharpoons H_{ads}$).

Heyrovsky /Tafel reactions: Excess adsorbed protons undergo the Heyrovsky reaction ($H_{ads} + H^+ + e^- \rightleftharpoons H_2$) or the Tafel reaction ($H_{ads} + H_{ads} \rightleftharpoons H_2$) resulting in H₂ gas evolution.

3) **Relationship between conductance and H₂ generation.** The increase in conductance is primarily driven by proton intercalation and the associated phase transition, which enhances carrier concentration and conductance. The generation of H₂ is a secondary effect, occurring due to the high proton flux and the catalytic activity of the MoO_x film under these conditions. The enhanced conductance facilitates the electrocatalytic reactions, leading to efficient H₂ production.

For revision, we have revised the main text and figure captions to explicitly state the causal relationship between proton intercalation, phase transition, and conductivity increase, and to distinguish this from the generation of hydrogen gas. Specifically, we have clarified that (Page 8, paragraph 1): “Note that the conductance increase is primarily driven by proton intercalation-induced phase transitions, with H₂ generation being a secondary catalytic effect at high Q_H due to excessive proton accumulation.”

5. On page 9, lines 10–12, the text seems to describe Supplementary Fig. S4 rather than Supplementary

Fig. S3. Please revise this section to reference the appropriate figure correctly.

Reply: Thanks for the comment. We have revised the corresponding statement from the supplementary Fig. S3 to supplementary Fig. S4 in Page 9, paragraph 2.

6. In Figure 4, the authors emulate the human memory process by using the output of the 1st ECRAM as the gate input for the 2nd ECRAM, effectively controlling the proton flux. The implementation of rank-order-coded spiking neural networks using this approach is highly impressive. However, the demonstration of the memory function using light in Figure 4a and 4b, described as being utilized for an organic photodetector, appears to be presented alongside the rank-order-coded spiking neural network results. Fig. 4a, b placement may give the impression that the light-induced memory function is directly related to the rank-order-coded network, potentially confusing readers.

Reply: Thank you for your insightful comment. We understand that the placement of Figure 4a and 4b may have given the impression that the light-induced memory function is directly related to the rank-order-coded spiking neural network (SNN). We appreciate the opportunity to clarify this point.

In order to avoid confusion for readers and emphasize the role of this section, we have split the Figure 4 (raw Fig. 4a-d to an individual Figure 4 as the **Fig. R6** and raw Fig. 4e-o to Figure 5) to more clearly distinguish between the light-induced memory function and the SNN implementation. Moreover, we add the data of the response time on light pulse as Figure 4c,d, and the data of the G of 2nd ECRAM vs. G of 1st ECRAM as Figure 4g,h. In Page 12 paragraph 3: “When the G of the 1st ECRAM increases from 0.36 μ S to 2.49 μ S, the conductance of the 2nd ECRAM varies linearly at different pulse number (**Fig. 4g**), and the average change in G (ΔG), stimulated by identical pulses, escalates from 0.017 μ S to 0.083 μ S (**Fig. 4h**).” In Page 12 paragraph 2: “The fast phase transition ensures the erasing and writing speed is comparable with the photodetector (**Fig. 4c,d, supplementary Fig. S15-17**).”

Figure R6. Integration of MoO_x ECRAMs for light and electric response. **a**, Circuit schematic of ECRAM with photodetector at gate electrode. **b**, Programming conductance plotted against number of light pulses. **c,d**, The rise time and fall time in the potentiation and depression. **e**, Circuit schematic of two connected ECRAMs. **f**, Programming conductance of 2nd ECRAM across various conductance levels of 1st ECRAM. **g**, The conductance of 2nd ECRAM as a function of the conductance of 1st ECRAM. **h**, The average change of conductance ΔG of 2nd ECRAM as a function of the conductance of 1st ECRAM.

Reference

- 1 Yang, C. S., Shang, D. S., Liu, N., Shi, G., Shen, X., Yu, R. C., Li, Y. Q. & Sun, Y. A Synaptic Transistor based on Quasi-2D Molybdenum Oxide. *Advanced Materials* **29**, 1700906 (2017).
- 2 Yang, C. S., Shang, D. S., Liu, N., Fuller, E. J., Agrawal, S., Talin, A. A., Li, Y. Q., Shen, B. G. & Sun, Y. All-Solid-State Synaptic Transistor with Ultralow Conductance for Neuromorphic Computing. *Advanced Functional Materials* **28**, 1804170 (2018).
- 3 Ge, C., Li, G., Zhou, Q.-l., Du, J.-y., Guo, E.-j., He, M., Wang, C., Yang, G.-z. & Jin, K.-j. Gating-induced reversible HxVO₂ phase transformations for neuromorphic computing. *Nano Energy* **67**, 104268 (2020).
- 4 Li, Y., Lu, J., Shang, D., Liu, Q., Wu, S., Wu, Z., Zhang, X., Yang, J., Wang, Z., Lv, H. & Liu, M. Oxide-Based Electrolyte-Gated Transistors for Spatiotemporal Information Processing. *Advanced Materials* **32**, 2003018 (2020).
- 5 Yao, X., Klyukin, K., Lu, W., Onen, M., Ryu, S., Kim, D., Emond, N., Waluyo, I., Hunt, A., Del Alamo, J. A., Li, J. & Yildiz, B. Protonic solid-state electrochemical synapse for physical neural networks. *Nat Commun* **11**, 3134

(2020).

- 6 Nikam, R. D., Kwak, M. & Hwang, H. All-Solid-State Oxygen Ion Electrochemical Random-Access Memory for Neuromorphic Computing. *Advanced Electronic Materials* **7**, 2100142 (2021).
- 7 Yang, J., Xiao, X., Chen, P., Zhu, K., Cheng, K., Ye, K., Wang, G., Cao, D. & Yan, J. Creating oxygen-vacancies in MoO₃-nanobelts toward high volumetric energy-density asymmetric supercapacitors with long lifespan. *Nano Energy* **58**, 455-465 (2019).
- 8 Cheng, C., Wang, A., Humayun, M. & Wang, C. Recent advances of oxygen vacancies in MoO₃: preparation and roles. *Chemical Engineering Journal* **498**, 155246 (2024).
- 9 Wang, M., Cai, S., Pan, C., Wang, C., Lian, X., Zhuo, Y., Xu, K., Cao, T., Pan, X., Wang, B., Liang, S.-J., Yang, J. J., Wang, P. & Miao, F. Robust memristors based on layered two-dimensional materials. *Nature Electronics* **1**, 130-136 (2018).
- 10 Jiang, J., Guo, J., Wan, X., Yang, Y., Xie, H., Niu, D., Yang, J., He, J., Gao, Y. & Wan, Q. 2D MoS₂ Neuromorphic Devices for Brain-Like Computational Systems. *Small* **13**, 1700933 (2017).
- 11 Zhu, X., Li, D., Liang, X. & Lu, W. D. Ionic modulation and ionic coupling effects in MoS₂ devices for neuromorphic computing. *Nature Materials* **18**, 141-148 (2018).
- 12 Wang, B., Wang, X., Wang, E., Li, C., Peng, R., Wu, Y., Xin, Z., Sun, Y., Guo, J., Fan, S., Wang, C., Tang, J. & Liu, K. Monolayer MoS₂ Synaptic Transistors for High-Temperature Neuromorphic Applications. *Nano Letters* **21**, 10400-10408 (2021).
- 13 Yang, J. T., Ge, C., Du, J. Y., Huang, H. Y., He, M., Wang, C., Lu, H. B., Yang, G. Z. & Jin, K. J. Artificial Synapses Emulated by an Electrolyte-Gated Tungsten-Oxide Transistor. *Advanced Materials* **30**, 1801548 (2018).
- 14 Tang, J., Bishop, D. M., Kim, S., Copel, M., Gokmen, T., Todorov, T. K., Shin, S., Lee, K.-T., Solomon, P. M., Chan, K. K. H., Haensch, W. E. & Rozen, J. in *2018 IEEE International Electron Devices Meeting (IEDM)*. 13.11.11-13.11.14.
- 15 Go, J., Kim, Y., Kwak, M., Song, J., Chekol, S. A., Kwon, J.-D. & Hwang, H. W/WO_{3-x} based three-terminal synapse device with linear conductance change and high on/off ratio for neuromorphic application. *Applied Physics Express* **12**, 026503 (2019).
- 16 Kim, S., Todorov, T., Onen, M., Gokmen, T., Bishop, D., Solomon, P., Lee, K. T., Copel, M., Farmer, D. B., Ott, J. A., Ando, T., Miyazoe, H., Narayanan, V. & Rozen, J. in *2019 IEEE International Electron Devices Meeting (IEDM)*. 35.37.31-35.37.34.
- 17 Lee, J., Nikam, R. D., Lim, S., Kwak, M. & Hwang, H. Excellent synaptic behavior of lithium-based nano-ionic transistor based on optimal WO_{2.7} stoichiometry with high ion diffusivity. *Nanotechnology* **31**, 235203 (2020).
- 18 Huang, F., Liu, H., Li, X. & Wang, S. Enhancing hole injection by processing ITO through MoO₃ and self-assembled monolayer hybrid modification for solution-processed hole transport layer-free OLEDs. *Chemical Engineering Journal* **427**, 131356 (2022).
- 19 Meng, X., Ji, W., Hua, J., Yu, Z., Zhang, Y., Li, H. & Zhao, J. Efficient, air-stable quantum dots light-emitting

- devices with MoO₃ modifying the anode. *Journal of Luminescence* **143**, 442-446 (2013).
- 20 Liang, D., Xue, X., Peng, J. & Ji, W. Perovskite light-emitting diodes with solution-processed MoO₃ films as the hole-transport layers. *Journal of Luminescence* **256**, 119621 (2023).
- 21 Brinkmann, K. O., Becker, T., Zimmermann, F., Kreusel, C., Gahlmann, T., Theisen, M., Haeger, T., Olthof, S., Tüchtmantel, C., Günster, M., Maschwitz, T., Göbelsmann, F., Koch, C., Hertel, D., Caprioglio, P., Peña-Camargo, F., Perdígón-Toro, L., Al-Ashouri, A., Merten, L., Hinderhofer, A., Gomell, L., Zhang, S., Schreiber, F., Albrecht, S., Meerholz, K., Neher, D., Stolterfoht, M. & Riedl, T. Perovskite–organic tandem solar cells with indium oxide interconnect. *Nature* **604**, 280-286 (2022).
- 22 Vibavakumar, S., Nisha, K. D., Harish, S., Archana, J. & Navaneethan, M. Synergistic effect of MoO₃/MoS₂ in improving the electrocatalytic performance of counter electrode for enhanced efficiency in dye-sensitized solar cells. *Materials Science in Semiconductor Processing* **161**, 107431 (2023).
- 23 Schmidt, H., Flügge, H., Winkler, T., Bülow, T., Riedl, T. & Kowalsky, W. Efficient semitransparent inverted organic solar cells with indium tin oxide top electrode. *Applied Physics Letters* **94**, 243302 (2009).
- 24 Kishi, M., Kubo, Y., Ishikawa, R., Shirai, H. & Ueno, K. Efficient Organic Photovoltaic Cells Using MoO₃ Hole-Transporting Layers Prepared by Simple Spin-Cast of Its Dispersion Solution in Methanol. *Japanese Journal of Applied Physics* **52**, 020202 (2013).
- 25 Lee, D. J., Kumar, G. M., Kim, Y., Yang, W., Kim, D. Y., Kang, T. W. & Ilanchezhian, P. Hybrid CsPbBr₃ quantum dots decorated two dimensional MoO₃ nanosheets photodetectors with enhanced performance. *Journal of Materials Research and Technology* **18**, 4946-4955 (2022).
- 26 Bai, Y., Liu, X., Chen, L., Khizar ul, H., Khan, M. A., Zhu, W. Q., Jiang, X. Y. & Zhang, Z. L. Organic thin-film field-effect transistors with MoO₃/Al electrode and OTS/SiO₂ bilayer gate insulator. *Microelectronics Journal* **38**, 1185-1190 (2007).
- 27 Zhang, Q., Ou, Q., Hu, G., Liu, J., Dai, Z., Fuhrer, M. S., Bao, Q. & Qiu, C.-W. Hybridized Hyperbolic Surface Phonon Polaritons at α -MoO₃ and Polar Dielectric Interfaces. *Nano Letters* **21**, 3112-3119 (2021).
- 28 Zhang, L., Jin, Z. & Tsubaki, N. Activating and optimizing the MoS₂@MoO₃ S-scheme heterojunction catalyst through interface engineering to form a sulfur-rich surface for photocatalyst hydrogen evolution. *Chemical Engineering Journal* **438**, 135238 (2022).
- 29 Al-Naggar, A. H., Shinde, N. M., Kim, J.-S. & Mane, R. S. Water splitting performance of metal and non-metal-doped transition metal oxide electrocatalysts. *Coordination Chemistry Reviews* **474**, 214864 (2023).
- 30 Avani, A. V. & Anila, E. I. Recent advances of MoO₃ based materials in energy catalysis: Applications in hydrogen evolution and oxygen evolution reactions. *International Journal of Hydrogen Energy* **47**, 20475-20493 (2022).
- 31 Zhang, Y., Fu, Q., Song, B. & Xu, P. Regulation Strategy of Transition Metal Oxide-Based Electrocatalysts for Enhanced Oxygen Evolution Reaction. *Accounts of Materials Research* **3**, 1088-1100 (2022).
- 32 Kwak, H., Kim, N., Jeon, S., Kim, S. & Woo, J. Electrochemical random-access memory: recent advances in materials, devices, and systems towards neuromorphic computing. *Nano Convergence* **11**, 9 (2024).

- 33 Cui, J., Liu, H. & Cao, Q. Prospects and challenges of electrochemical random-access memory for deep-learning accelerators. *Current Opinion in Solid State and Materials Science* **32**, 101187 (2024).
- 34 Tan, L., Liu, W., Feng, Y., Yao, Y., Zhan, C., Pan, J., Li, H., Zhang, L. & Xiong, L. Unlocking the Critical Role of Anion and Oxygen Vacancies in MoO_{3-x} Nanobelts for High-Performance Proton Storage. *ACS Applied Nano Materials* **7**, 19377-19385 (2024).
- 35 Luo, Z., Miao, R., Huan, T. D., Mosa, I. M., Poyraz, A. S., Zhong, W., Cloud, J. E., Kriz, D. A., Thanneeru, S., He, J., Zhang, Y., Ramprasad, R. & Suib, S. L. Mesoporous MoO₃- Material as an Efficient Electrocatalyst for Hydrogen Evolution Reactions. *Advanced Energy Materials* **6**, 1600528 (2016).
- 36 Seh, Z. W., Kibsgaard, J., Dickens, C. F., Chorkendorff, I., Nørskov, J. K. & Jaramillo, T. F. Combining theory and experiment in electrocatalysis: Insights into materials design. *Science* **355**, eaad4998 (2017).
- 37 Yu, Y., Nam, G.-H., He, Q., Wu, X.-J., Zhang, K., Yang, Z., Chen, J., Ma, Q., Zhao, M., Liu, Z., Ran, F.-R., Wang, X., Li, H., Huang, X., Li, B., Xiong, Q., Zhang, Q., Liu, Z., Gu, L., Du, Y., Huang, W. & Zhang, H. High phase-purity 1T'-MoS₂- and 1T'-MoSe₂-layered crystals. *Nature Chemistry* **10**, 638-643 (2018).
- 38 Yang, J., Shao, Q., Huang, B., Sun, M. & Huang, X. pH-Universal Water Splitting Catalyst: Ru-Ni Nanosheet Assemblies. *iScience* **11**, 492-504 (2019).
- 39 Feng, H., Xu, Z., Ren, L., Liu, C., Zhuang, J., Hu, Z., Xu, X., Chen, J., Wang, J., Hao, W., Du, Y. & Dou, S. X. Activating Titania for Efficient Electrocatalysis by Vacancy Engineering. *ACS Catalysis* **8**, 4288-4293 (2018).
- 40 Li, Y., Fuller, E. J., Asapu, S., Agarwal, S., Kurita, T., Yang, J. J. & Talin, A. A. Low-Voltage, CMOS-Free Synaptic Memory Based on LiXTiO₂ Redox Transistors. *ACS Applied Materials & Interfaces* **11**, 38982-38992 (2019).
- 41 Shao, Z., Cao, X., Luo, H. & Jin, P. Recent progress in the phase-transition mechanism and modulation of vanadium dioxide materials. *NPG Asia Materials* **10**, 581-605 (2018).
- 42 Zhou, W., Li, X., Li, X., Shao, J., Yang, H., Chai, X., Hu, Q. & He, C. Crafting amorphous VO₂-crystalline NiS₂ heterostructures as bifunctional electrocatalysts for efficient water splitting: The different cocatalytic function of VO₂. *Chemical Engineering Journal* **470**, 144146 (2023).
- 43 Tang, J., Bishop, D., Kim, S., Copel, M., Gokmen, T., Todorov, T., Shin, S., Lee, K. T., Solomon, P., Chan, K., Haensch, W. & Rozen, J. in *2018 IEEE International Electron Devices Meeting (IEDM)*. 13.11.11-13.11.14.
- 44 Chen, J., Chen, C., Qin, M., Li, B., Lin, B., Mao, Q., Yang, H., Liu, B. & Wang, Y. Reversible hydrogen spillover in Ru-WO_{3-x} enhances hydrogen evolution activity in neutral pH water splitting. *Nature Communications* **13**, 5382 (2022).
- 45 Mo, X., Wang, J., Wang, Z. & Wang, S. The application of polypyrrole fibrils in hydrogen evolution reaction. *Synthetic Metals* **142**, 217-221 (2004).
- 46 Gao, D., Zhao, S., Huang, Y., Wu, X., Li, R., Ding, Y., Jiang, Q., Zhao, Y., Wang, F. & Zhang, R. A Facile Electrochemical Strategy for Achieving a High-Conductivity Polypyrrole Derivative with Intrinsic Metallic Transport as a High-Performance Electrochromic Conducting Polymer Film. *Nano Letters* **24**, 14854-14861 (2024).

- 47 Cho, K. G., Adrahtas, D. Z., Lee, K. H. & Frisbie, C. D. Sub-Band Filling and Hole Transport in Polythiophene-Based Electrolyte-Gated Transistors: Effect of Side-Chain Length and Density. *Advanced Functional Materials* **33**, 2303700 (2023).
- 48 Ng, C. H., Winther-Jensen, O., Ohlin, C. A. & Winther-Jensen, B. Enhanced catalytic activity towards hydrogen evolution on polythiophene via microstructural changes. *International Journal of Hydrogen Energy* **42**, 886-894 (2017).
- 49 Shakya, J., Kang, M.-A., Li, J., VahidMohammadi, A., Tian, W., Zeglio, E. & Hamed, M. M. 2D MXene electrochemical transistors. *Nanoscale* **16**, 2883-2893 (2024).
- 50 Tekalgne, M. A., Do, H. H., Nguyen, T. V., Le, Q. V., Hong, S. H., Ahn, S. H. & Kim, S. Y. MXene Hybrid Nanosheet of WS₂/Ti₃C₂ for Electrocatalytic Hydrogen Evolution Reaction. *ACS Omega* **8**, 41802-41808 (2023).
- 51 Hu, X., Gao, Y., Luo, X., Xiong, J., Chen, P. & Wang, B. Insight into the intrinsic activity of various transition metal sulfides for efficient hydrogen evolution reaction. *Nanoscale* **16**, 4909-4918 (2024).
- 52 Endrődi, B., Stojanovic, A., Cuartero, M., Simic, N., Wildlock, M., de Marco, R., Crespo, G. A. & Cornell, A. Selective Hydrogen Evolution on Manganese Oxide Coated Electrodes: New Cathodes for Sodium Chlorate Production. *ACS Sustainable Chemistry & Engineering* **7**, 12170-12178 (2019).
- 53 Zhao, X., Yin, F., He, X., Chen, B. & Li, G. Enhancing hydrogen evolution reaction activity on cobalt oxide in alkaline electrolyte by doping inactive rare-earth metal. *Electrochimica Acta* **363**, 137230 (2020).
- 54 Wu, Z., Liao, T., Wang, S., Mudiyansele, J. A., Micallef, A. S., Li, W., O'Mullane, A. P., Yang, J., Luo, W., Ostrikov, K., Gu, Y. & Sun, Z. Conversion of Catalytically Inert 2D Bismuth Oxide Nanosheets for Effective Electrochemical Hydrogen Evolution Reaction Catalysis via Oxygen Vacancy Concentration Modulation. *Nano-Micro Letters* **14**, 90 (2022).
- 55 Yu, Y., Li, G., Xiao, Y., Chen, C., Bai, Y., Wang, T., Li, J., Hua, Y., Wu, D., Rao, P., Deng, P., Tian, X. & Yuan, Y. Iridium-based electrocatalysts for acidic oxygen evolution reaction. *Journal of Energy Chemistry* **103**, 200-224 (2025).
- 56 Gao, B., Jalem, R., Ma, Y. & Tateyama, Y. Li⁺ Transport Mechanism at the Heterogeneous Cathode/Solid Electrolyte Interface in an All-Solid-State Battery via the First-Principles Structure Prediction Scheme. *Chemistry of Materials* **32**, 85-96 (2019).
- 57 Cheng, Z., Liu, M., Ganapathy, S., Li, C., Li, Z., Zhang, X., He, P., Zhou, H. & Wagemaker, M. Revealing the Impact of Space-Charge Layers on the Li-Ion Transport in All-Solid-State Batteries. *Joule* **4**, 1311-1323 (2020).